# An alternative EGFR activation by patient-derived R252C mutation promotes cancer progression

Yajuan Zhang [1,2,10], Qizhen Fei[3,10], Yan Li [4,10], Siyao Wang[2,10], Tong Rong[1], Xueyuan Wu[5], Hong Gao[2], Chen Chen[6,7], Dong Gao [2], Yun Zhao [2], Guohui Li [4], Huiying Chu [4] ✉, Wenfeng Li [6] ✉ & Weiwei Yang [2,8,9] ✉

Mutations in the extracellular or intracellular domains of epidermal growth factor receptor (EGFR) are implicated in the development of various cancers. While the intracellular mutations of EGFR have been extensively studied, the function of extracellular mutations remains poorly understood. In this study, we identify an EGFR mutant (EGFR R252C) in a patient with multifocal lung cancer and glioma, in which arginine (R) 252 is mutated to cysteine (C) in the EGFR extracellular domain. This mutation promotes C252-C252 disulfide-mediated EGFR dimerization and induces a conformational change of EGFR, leading to absent autophosphorylation and enhanced direct interaction between EGFR and extracellular signal-regulated protein kinase 1/2 (ERK1/2). Importantly, EGFR directly phosphorylates ERK1/2 at threonine (T) 202 / tyrosine (Y) 204 and activates ERK1/2, thereby promoting tumor cell proliferation and tumor growth in vivo. Afatinib, a second-generation EGFR tyrosine kinase inhibitor, effectively suppresses primary tumor growth and extends progression-free survival in the patient with multifocal lung cancer and glioma driven by EGFR R252C. Our finding elucidates the activation mechanism of this extracellular EGFR mutation and demonstrates the efficacy of afatinib in treating lung cancer or glioma patients with this variant.

Epidermal growth factor receptor (EGFR) is a transmembrane receptor tyrosine kinase protein that has been implicated in tumor cell proliferation, angiogenesis, tumor invasion, metastasis, and inhibition of apoptosis. EGFR mutations are observed in approximately 5% of all cancer patients, 14% of non-small cell lung cancer (NSCLC) patients, and 26% of glioma patients[1–3]. In lung cancer, EGFR mutations predominantly occur in the intracellular kinase domain (KD); these mutations enhance receptor activation by stabilizing the active

[1]Shanghai Institute of Thoracic Oncology, Shanghai Chest Hospital, Shanghai Jiao Tong University School of Medicine, Shanghai, China. [2]Key Laboratory of Multi-cell Systems, Shanghai Key Laboratory of Molecular Andrology, CAS Center for Excellence in Molecular Cell Science, Shanghai Institute of Biochemistry and Cell Biology, University of Chinese Academy of Sciences, Chinese Academy of Sciences, Shanghai, China. [3]Department of Thoracic Surgery, Shanghai Chest Hospital, Shanghai Jiao Tong University School of Medicine, Shanghai, China. [4]Interdisciplinary Research Center for Biology and Chemistry, Liaoning Normal University, Dalian, Liaoning, China. [5]Department of Oncology, Renji Hospital, School of Medicine, Shanghai Jiao Tong University, Shanghai, China. [6]Department of Radiation Oncology, First Affiliated Hospital, Wenzhou Medical University, Wenzhou, Zhejiang, China. [7]Zhejiang Key Laboratory of Intelligent Cancer Biomarker Discovery and Translation, First Affiliated Hospital, Wenzhou Medical University, Wenzhou, Zhejiang, China. [8]Key Laboratory of Systems Health Science of Zhejiang Province, School of Life Science, Hangzhou Institute for Advanced Study, University of Chinese Academy of Sciences, Hangzhou, Zhejiang, China. [9]Shanghai Academy of Natural Sciences (SANS), Shanghai, China. [10]These authors contributed equally: Yajuan Zhang, Qizhen Fei, Yan Li, Siyao Wang. ✉e-mail: chuhy2009@dicp.ac.cn; liwenfeng@wmu.edu.cn; wyang@sibcb.ac.cn

conformation of the KD, thereby facilitating the dimerization process. While in glioma, EGFR mutations mainly occur in the extracellular domain (ECD)[1,4]. EGFRvIII, the most commonly identified mutation in glioma, features an in-frame deletion between residues 6 and 273 within the ECD, which impedes EGF binding to the receptor while maintaining its kinase activity[5]. Furthermore, point mutations in the ECD of EGFR, such as arginine (R) 108 to lysine (K), alanine (A) 289 to valine (V)/aspartate (D)/threonine (T), glycine (G) 598 to V, and others, have been identified in glioma samples[6]. Mutations in both the EGFR KD and ECD are common in cancer patients and hold significant importance. However, compared to the EGFR KD, research on mutations in the ECD remains relatively scarce. Therefore, further studies on EGFR ECD mutations are urgently needed.

Given the prevalence of EGFR mutations observed in numerous cancer patients, several generations of EGFR TKIs have been developed, with ongoing research efforts persisting in this field. EGFR mutations occurring at different positions within the KD exhibit varying sensitivities to different generations of EGFR TKIs. For instance, point mutations at exon 21 (leucine 858 to arginine) show favorable responses to both first- and third-generation EGFR TKIs. However, mutations such as G719X, S768I, and L861Q demonstrate greater sensitivity to second-generation TKIs[1]. In contrast to the well-established research and therapeutic advancements for EGFR KD mutations, progress for EGFR ECD mutations has lagged behind. Although EGFR ECD mutations also alter the structure and function of EGFR, leading to aberrant activation of the EGFR signaling pathway, existing EGFR TKIs demonstrate limited efficacy against these mutations. This limited response may be attributed to the conformational changes or differences in the signaling mechanisms induced by ECD mutations[4,6]. Currently, no approved treatment strategies exist for cancers driven by EGFR mutations within ECD[1]. Therefore, investigating novel therapies for these mutations is not only urgent but also essential. Only through further in-depth exploration of the mechanisms and biological characteristics of ECD mutations can we develop new treatments that improve patient outcomes and provide effective treatment options for a broader population of EGFR-mutant patients.

In this study, we identified an EGFR mutation within the ECD from a patient with multifocal lung cancer and glioma. We conducted a comprehensive analysis of the activation mechanisms and functional properties associated with this mutation and proposed that second-generation EGFR-TKI afatinib effectively inhibited tumor progression driven by this alteration. Notably, treatment with afatinib led to a substantial reduction in tumor burden and prolonged progression-free survival in the patient with multifocal lung cancer and glioma harboring this mutation.

## Results
### EGFR R252C promotes tumor cell proliferation and tumor growth in vivo
We admitted a patient with malignant tumor lesions detected on both lung CT and brain MRI (Fig. 1a). The patient was initially treated with four cycles of platinum-based chemotherapy combined with pemetrexed, with CT imaging demonstrating a partial response (PR). To further suppress tumor progression, a combination of chemotherapy and bevacizumab was administered, but the patient experienced tumor enlargement. Consequently, KEYTRUDA immunotherapy was then initiated, but the patient showed no response. The treatment timeline of this patient was shown in Fig. 1b. To explore potential treatment options for the patient, we investigated the underlying causes of the disease. Circulating tumor DNA (ctDNA) sequencing of the patient's peripheral blood was conducted to identify clinically actionable mutations. No pathogenic variants were detected in common hotspots (NRAS Q61/G13/G12, PIK3CA H1047, EGFR G719/L858, BRAF V600, or KRAS Q61/G13/G12). Notably, comprehensive genomic profiling uncovered a rare EGFR R252C mutation. This suggests that

the EGFR R252C mutation may have oncogenic activity and induce tumor growth in the patient.

Considering the presence of tumor lesions in both the lungs and brain of the patient, we searched the Cancer Genome Atlas (TCGA) database for the EGFR R252C mutation. We found that this mutation occurs in 0.4 % of glioma patients, but it was not detected in lung cancer patients. However, in an unpublished database from China, among 4000 lung cancer patients, we identified one patient harboring this mutation. To further confirm the function of EGFR R252C, we adopted CRISPR-Cas9 technique to generate EGFR R252C knock-in U87 and U251 human glioblastoma cells and H1299 human non-small cell lung cancer cells. Successful gene knock-in was validated by genetic sequencing (Fig. 1c). EGFR R252C did not obviously influence EGFR expression (Fig. 1d, e). As shown in Fig. 1f, g, EGFR R252C expression markedly increased the proliferation and colony formation in these tumor cells compared to EGFR WT.

To validate our findings in a clinically relevant GBM model, we utilized the patient-derived GSC387 glioma stem cell line, which maintains key molecular and phenotypic characteristics of primary glioblastoma[7]. Using lentiviral-mediated gene delivery, we established GSC387 cell lines stably expressing either HA-tagged EGFR WT or the R252C mutant. The expression of EGFR R252C significantly enhanced GSC387 proliferation compared to EGFR WT (Supplementary Fig. 1a,b).

Furthermore, we intracranially injected luciferase-expressing U87 cells with or without EGFR R252C mutation into randomized athymic nude mice. Bioluminescence imaging of mice showed that mice bearing intracranial tumors with the EGFR R252C mutation demonstrated a more rapid growth rate compared to those implanted with tumor cells expressing EGFR WT (Fig. 1h). Additionally, we injected luciferase-expressing H1299 cells harboring EGFR WT or R252C into the left lung of athymic nude mice via orthotopic lung injection. Similarly, the results indicated that the growth of lung tumors in mice with H1299 cells expressing EGFR mutation exhibited a faster growth rate compared to those with EGFR WT cells (Fig. 1i). In summary, EGFR R252C mutation promotes cell proliferation and tumor growth in vivo.

### EGFR R252C phosphorylates and activates ERK1/2
To explore how EGFR R252C promotes tumor cell proliferation, we first examined the activation of EGFR and its downstream signaling pathways in U87, U251, and H1299 cells expressing EGFR WT or R252C. Strikingly, EGFR R252C expression did not influence the levels of phosphorylated (p) EGFR tyrosine (Y) 1068 (EGFR pY1068), phosphorylated MEK serine (S) 217/221 (MEK pS217/221), phosphorylated AKT threonine (T) 308 (AKT pT308) and phosphorylated STAT3 tyrosine 705 (STAT3 pY705), while it markedly increased the levels of phosphorylated ERK threonine 202 and tyrosine 204 (ERK1/2 pT202/Y204) (Fig. 2a). Then, we wondered whether the kinase activity of EGFR R252C is required for ERK1/2 activation. We incubated tumor cells expressing EGFR WT or R252C with afatinib, the EGFR inhibitor, which showed that EGFR R252C-activated ERK1/2 activation was suppressed by afatinib treatment (Fig. 2b). A similar result was obtained in GSC387 cells (Supplementary Fig. 2a).

ERK1/2 are the final components of the mitogen-activated protein kinase (MAPK) phosphorylation cascade, serving as a crucial module in various signaling pathways that shape cell behavior and fate, including cell proliferation[8]. To examine whether EGFR R252C enhanced cell proliferation through ERK1/2, we treated U87, U251, and H1299 cells expressing EGFR WT or R252C with ERK1/2 inhibitor, ravoxertinib. ERK1/2 inhibition abrogated EGFR R252C-promoted tumor cell proliferation (Fig. 2c), demonstrating that ERK1/2 activation serves as a critical downstream effector of EGFR R252C oncogenic signaling.

ERK1/2 activation is initiated through a sequential phosphorylation cascade involving receptor tyrosine kinase (RTK)-mediated RAS activation, subsequent RAF kinase recruitment, and ultimately MEK-dependent dual phosphorylation of ERK1/2 at their activation loop[8]. To

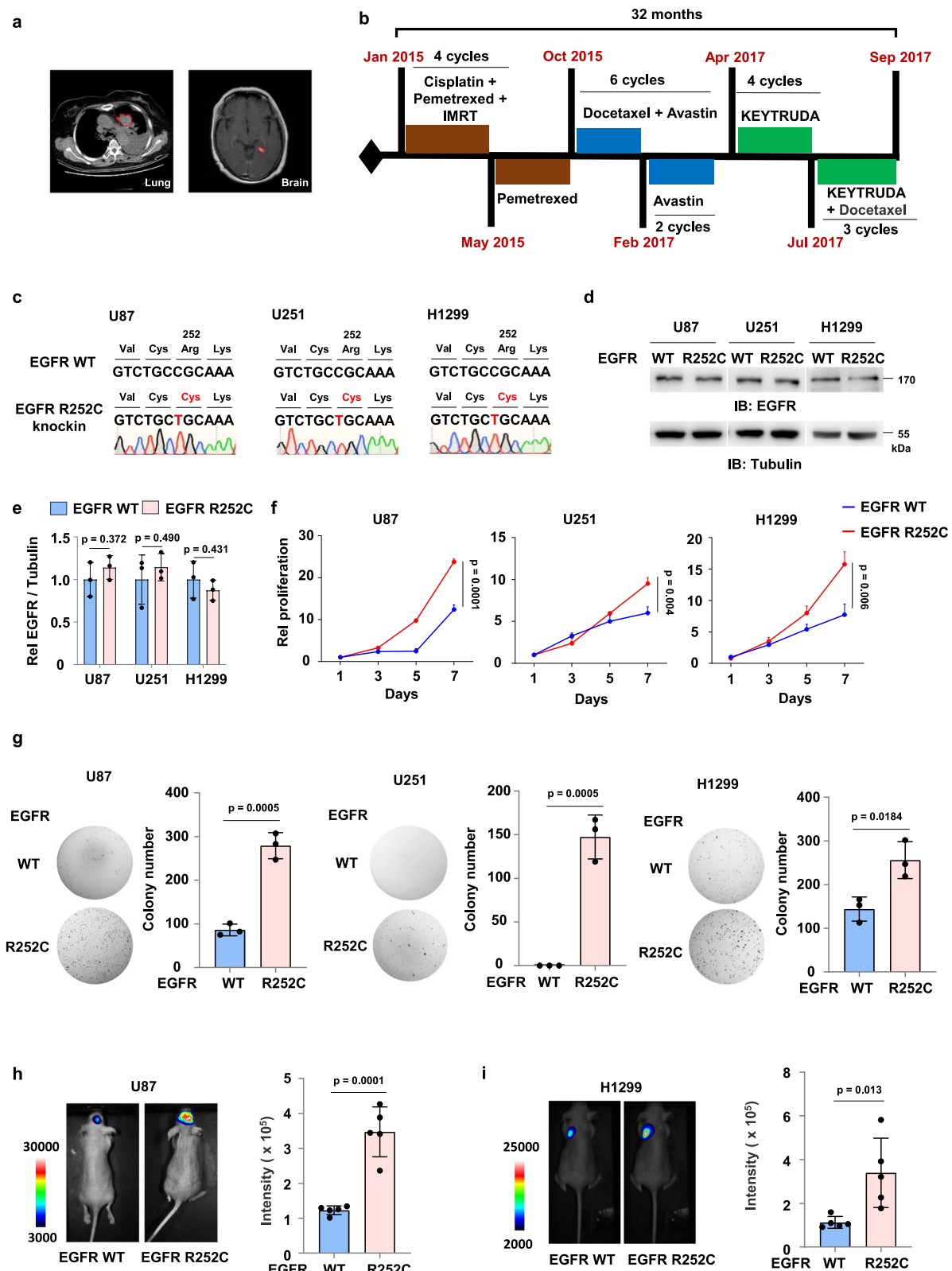

investigate whether EGFR R252C-induced ERK1/2 activation is dependent on the canonical RAS/RAF/MEK pathway, we treated U251 cells harboring either EGFR WT or EGFR R252C with RAS inhibitor (tipifarnib), RAF inhibitor (dabrafenib), or MEK inhibitor (trametinib), respectively. As shown in Supplementary Fig. 2b–d, inhibition of RAS, RAF, or MEK failed to block EGFR R252C-induced ERK1/2 activation. The efficacy of the above inhibitors was assessed under EGF

stimulation (Supplementary Fig. 2e). Above results indicated that EGFR R252C-induced ERK1/2 activation is independent of the canonical RAS/RAF/MEK pathway. This conclusion was further validated by MEK1/2 depletion experiments in both U251 and H1299 cell lines (Supplementary Fig. 2f, g).

To elucidate how EGFR R252C activates ERK1/2 independent of MEK1/2 activation, we performed co-immunoprecipitation (co-IP)

**Fig. 1 | EGFR R252C promotes tumor cell proliferation and tumor growth in vivo. a** Chest CT and brain MRI images of a patient with multifocal lung cancer and glioma. **b** Treatment timeline of the patient. **c** DNA sequencing of genomic DNA from U87, U251, or H1299 cells with or without EGFR R252C knock-in. **d, e** Immunoblotting assay was performed with U87, U251, and H1299 cells with or without EGFR R252C knock-in. Tubulin was used to determine equivalent loading. IB, immunoblotting. **d** Representative images of immunoblotting were shown. **e** Semi-quantitative scoring was carried out. The EGFR/Tubulin ratio was quantified and normalized to the EGFR WT cells. Rel, relative. **f** U87, U251 or H1299 cells harboring EGFR WT or R252C mutation were seeded in 96-well plates. Cell proliferation assay was performed. Relative cell proliferation of U87, U251, and H1299 cells expressing EGFR WT or R252C was normalized to the day 1 value. **g** Colony formation assay in U87, U251 or H1299 cells with or without EGFR R252C mutation. A representative experiment is shown in the left panel. The number of colonies was calculated (right panel). **h** Luciferase-expressing U87 cells with or without EGFR R252C knock-in were intracranially injected into randomized athymic nude mice (five mice per group). 49 days after implantation, bioluminescence imaging of these mice was carried out to examine tumor growth. Data represent the mean ± s.d. of five mice. **i** Luciferase-expressing H1299 cells with or without EGFR R252C knock-in were injected into the left lung of randomized athymic nude mice (five mice per group). 35 days after implantation, bioluminescence imaging of mice was carried out to examine tumor growth. Data represent the mean ± s.d. of five mice. **e, f, g** Data represent the mean ± s.d. of three biologically independent experiments. **e–i**, Unpaired, two-tailed t-test.

experiments in EGFR WT and EGFR R252C cells. We found that EGFR directly interacted with ERK1/2 and R252C enhanced the interaction between EGFR and ERK1/2 in U87, U251, or H1299 cells (Fig. 2d). It was further evidenced by the in vitro pulldown assay (Fig. 2e). To test whether EGFR R252C directly phosphorylates ERK1/2, we performed the in vitro kinase assays by mixing recombinant His-ERK1 or GST-ERK2 with recombinant Flag-EGFR WT or R252C. It was shown that EGFR directly phosphorylated ERK1/2, while more ERK1/2 were phosphorylated by EGFR R252C (Fig. 2f, g). Taken together, these results suggest that EGFR R252C directly phosphorylates T202/Y204 to activate ERK1/2, thereby promoting the proliferation of tumor cells.

## Structural basis of EGFR R252C-dependent ERK1/2 activation

Upon binding with its ligand, EGFR forms either homodimers or heterodimers, subsequently activating the intracellular kinase domain and initiating downstream signaling pathways[9]. To determine whether R252C affects ERK1/2 phosphorylation by affecting the dimerization of EGFR, we performed co-IP experiments and observed that EGFR R252C forms more dimers than EGFR WT (Fig. 3a).

Next, we investigated the structural basis of EGFR R252C-enhanced EGFR-ERK1/2 interaction by performing molecular dynamics (MD) simulation for the ligand-free EGFR R252C ECD (PDB code 3NJP[10]). The conformations sampled from independent trajectories based on the clustering analysis[11] showed that the ligand binding site gap between domain I and III still maintained in the ligand-free state of EGFR R252C (Supplementary Fig. 3a). The main reason for the binding site gap existing was that EGFR R252C formed the disulfide bond between the two monomers, which stabilized the dimeric conformation.

To confirm whether R252C mutation stabilizes a unique ECD dimer of EGFR and whether the dimer formation is dependent on the C252-C252 disulfide, we transfected H1299 cells with HA-tagged EGFR WT, R252C, or R252S. The cells were harvested and analyzed by non-reducing SDS-PAGE to preserve disulfide bonds. As shown in Supplementary Fig. 3b, R252C mutation indeed promoted the dimer formation of EGFR, while R252S mutation failed to do so. The result confirmed that R252C stabilizes a unique EGFR ECD dimer specifically through C252-C252 disulfide bond formation. This conclusion was further validated by glutaraldehyde-mediated crosslinking experiments (Supplementary Fig. 3c). Besides, we transfected H1299 cells with HA-tagged EGFR WT, R252C, or R252S. The cells were harvested and analyzed by SDS-PAGE. As shown in Supplementary Fig. 3d, compared to EGFR WT, R252C mutation enhanced ERK1/2 activation, while EGFR R252S mutation failed to do so. These results confirmed that R252C mutation promotes the dimerization of EGFR to activate ERK1/2, which is dependent on the C252-C252 disulfide bond.

Moreover, in ECD of EGFR R252C dimer, the distance between C terminal (Cα of residue 638, $d_{cc}$) of the two subunits was more similar to that of ligand-bound EGFR WT rather than ligand-free EGFR WT (Supplementary Fig. 3e). Meanwhile, the angle formed by I214 and P228 (in the same subunit) and P228 (in the other subunit) was also more similar to that of ligand-bound EGFR WT (Supplementary Fig. 3f). As shown in Supplementary Fig. 3a, the ligand-free EGFR R252C, like the ligand-bound EGFR WT, maintained a staggered conformation, which contrasted with the ligand-free EGFR WT that exhibited a flush conformation. Therefore, in the ligand-free EGFR R252C, the extracellular dimer presented a "V shape" and staggered conformation, which was similar to the extracellular dimer of the ligand-bound EGFR WT, but with a smaller $d_{cc}$.

The transmembrane (TM) segment links the extracellular and intracellular domains. Because of the $d_{cc}$ of the EGFR R252C extracellular dimer conformation was similar to that of the ligand-bound EGFR WT, the N-terminal TM dimer was more suitable for the EGFR R252C in the absence of a ligand. The GxxxG motifs often form the dimerization interfaces of the TM helices[12–14]. In EGFR R252C, the distance between the center of mass of the two motifs in helix was about 7.5 Å, which was consistent with the results of ligand-bound EGFR WT obtained by Arkhipov et al.[12] (6 Å), but was significantly different from the ligand-free EGFR WT conformation (4.5 Å) (Supplementary Fig. 3g). The contact probability map of ligand-free EGFR R252C showed that the TM dimer maintained N-terminal contact, similar to that of the ligand-bound EGFR WT (Supplementary Fig. 3h).

In intact EGFR, juxtamembrane (JM) dimers directly connected the TM dimers and the KD. It has been previously reported that the JM dimers form an antiparallel helix in the active state of EGFR[12]. We found that the JM dimer of EGFR R252C was also antiparallel helix and stabilized along the simulations (Supplementary Fig. 4a). Hence, the conformation of the extracellular domain, TM, and JM dimers of ligand-free EGFR mutant was similar to that of the ligand-bound EGFR WT.

Due to the smaller $d_{cc}$ of the EGFR R252C ECD and the antiparallel helix structure of the JM, the active site in the asymmetric KD was exposed. The final conformation was shown in Fig. 3b. The distribution of the center of mass between the two KDs in the simulated ECD-TM-JM-KD dimer of EGFR R252C showed that significantly larger than ligand-bound EGFR WT (Fig. 3c). The PCA results revealed that in the EGFR R252C, the two KD regions behave differently: the residues in one KD remain nearly stationary, while the residues in the upper and lower parts of the other KD region move in opposite directions, causing the KD to twist and exposing the active site (Supplementary Fig. 4b). Although the active site faced the interior of the cell in both the ligand-free EGFR R252C and the ligand-bound EGFR WT, the PCA results indicate that the orientation of the active site was altered in the ligand-free EGFR R252C, causing it to deviate from the other KD (Supplementary Fig. 4b).

To corroborate our simulation results, we performed a NanoBiT assay, where two non-fluorescent subunits of the luciferase enzyme (LgBiT and SmBiT) are fused to EGFR WT or R252C. The experimental data indicated that compared to EGFR WT treated with EGF, EGFR R252C exhibited reduced fluorescence intensity, which is likely due to the increased distance between the two KD dimers in the EGFR R252C (Fig. 3d). Due to the change in the active site orientation in the ligand-

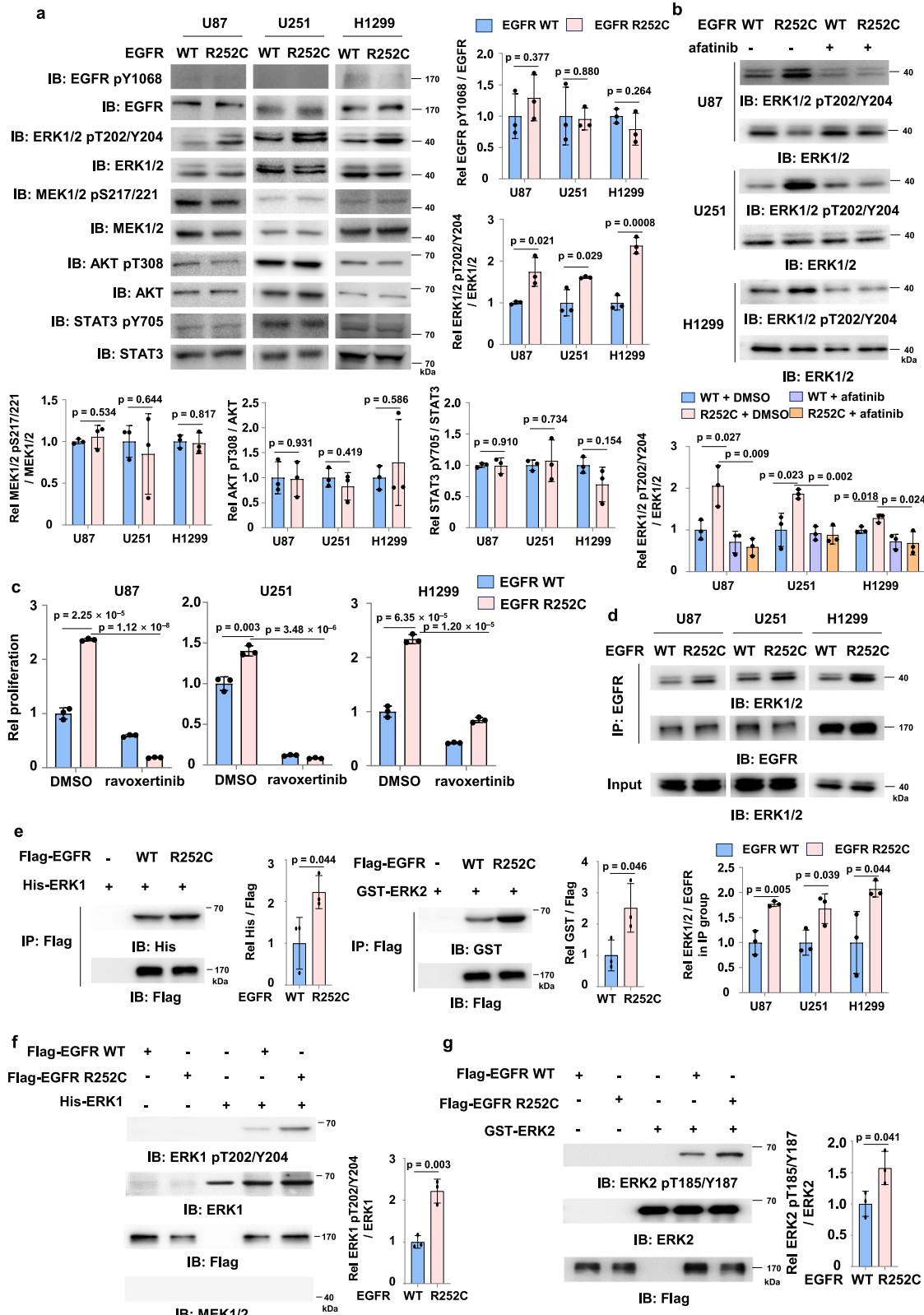

free EGFR R252C, the distance between the KD of the two subunits increases, resulting in absent autophosphorylation of EGFR R252C, compared to that in the activated state of EGFR WT (Supplementary Fig. 4e, f).

Thus, we evaluated the binding affinities between ERK1 and KD dimer of the ligand-free EGFR R252C and ligand-bound EGFR WT using MM-GBSA calculations. The stronger total binding free energies of KD

dimer of ligand-free EGFR R252C with ERK1 is −85.34 kcal/mol (Fig. 3e), which was significantly stronger than the ligand-bound EGFR WT with ERK1 (−48.53 kcal/mol). The dominant residues favoring for EGFR R252C KD dimer binding with ERK1 were R986, E804, E1004, L718, L1001, M1002, F795, R803, P919, S720 and K716 of ligand-free EGFR R252C (Fig. 3f). Through the analysis of native contact, combined with the possible electrostatic interaction analysis, the residues E922 and

**Fig. 2 | EGFR R252C phosphorylates and activates ERK1/2. a** U87, U251, and H1299 cells with or without EGFR R252C mutation were harvested and used for immunoblotting assay with indicated antibodies. The samples derive from the same experiment but different gels for phospho-EGFR, EGFR, another for phospho-ERK1/2, ERK1/2, another for phospho-MEK1/2, MEK1/2, another for phospho-AKT, AKT, and another for phospho-STAT3, STAT3 were processed in parallel. Semi-quantitative scoring was carried out. Ratios of phosphorylated protein level to total protein level were quantified for EGFR, ERK1/2, MEK1/2, AKT, and STAT3, and normalized to EGFR WT cells. **b** U87, U251, and H1299 cells, with or without EGFR R252C mutation, were treated with or without 10 μM afatinib for 6 h. Immuno-blotting assay was performed, and semi-quantitative scoring was carried out. The phosphorylated-ERK1/2 to ERK1/2 ratio was quantified and normalized to the vehicle-treated EGFR WT cells. **c** U87, U251, and H1299 cells, with or without EGFR R252C mutation were treated with or without 5 μM Ravoxertinib. Cell proliferation was assessed at day 7. Relative proliferation was calculated using vehicle-treated EGFR WT cells as the control. **d** Immunoprecipitation assay of U87, U251 and H1299 cells with or without EGFR R252C mutation. Semi-quantitative scoring was carried out, and the ratio of ERK1/2 to EGFR in IP group was quantified and normalized to the EGFR WT group. IP, immunoprecipitation. **e** In vitro pulldown assay was performed by mixing recombinant His-ERK1 or GST-ERK2 with Flag-EGFR WT or R252C. Semi-quantitative scoring was carried out. His/Flag and GST/Flag ratios were quantified and normalized to EGFR WT groups. **f, g** In vitro kinase assay was performed by mixing recombinant GST-ERK1 or GST-ERK2 and Flag-EGFR WT or R252C immunoprecipitated from HEK293T cells. Immunoblotting analyses were performed. Semi-quantitative scoring was carried out. The ratios of phosphorylated ERK1 to total ERK1 and phosphorylated ERK2 to total ERK2 were quantified and normalized to EGFR WT groups. **a–g** Data represent the mean ± s.d. of three biologically independent experiments (unpaired, two-tailed t-test).

I926 were also consider as the possible dominate residues of EGFR R252C.

To identify the dominant residues involved in the binding of EGFR R252C with ERK1/2, we generated mutations at R803, E804, P919, E922, I926, and R986 in the EGFR R252C and conducted immunoprecipitation assays, which identified P919 as a potential binding site between the EGFR R252C KD dimer and ERK1/2 (Fig. 3g and Supplementary Fig. 4g, h). Additionally, cell proliferation and soft agar colony formation assays demonstrated that the P919G mutation significantly abrogated the proliferative capacity of EGFR R252C cells (Fig. 3h, i and Supplementary Fig. 4i). Our experiments identified P919 as a critical binding site within the EGFR R252C for ERK1/2, providing evidence that it mediates the functional effects of EGFR R252C.

In summary, the simulated EGFR R252C model contains an active ligand-free extracellular dimer conformation, similar to the ligand-bound active EGFR WT extracellular dimer, including the N-terminal TM, antiparallel JM dimer, and active asymmetric KD dimer. The active site of the KD dimer was oriented toward the cell interior; however, the ligand-free EGFR R252C exhibits an increased distance between the KDs and an altered domain orientation. These changes are the primary reasons for the absent autophosphorylation levels observed in EGFR R252C compared to ligand-bound EGFR WT. The binding free energy between the EGFR mutant and ERK1 is stronger than that between EGFR WT and ERK1 due to the conformational change, leading to a dramatic increase in ERK1/2 phosphorylation levels.

### Afatinib inhibits EGFR R252C-driven tumor cell proliferation and tumor growth in vivo

Gefitinib and erlotinib are first-generation EGFR-TKIs that reversibly bind the ATP-pocket of EGFR, mainly targeting common mutations (ex19del/L858R). Afatinib, a second-generation inhibitor, irreversibly blocks EGFR with broader mutation coverage but higher toxicity. Osimertinib, the leading third-generation TKI, irreversibly inhibits both classic mutations and T790M resistance mutations while sparing wild-type EGFR, reducing side effects[15]. Studies have shown that mutations in different regions of EGFR elicit varying responses to EGFR TKIs. Mutations in the KD preferentially respond to the first-generation EGFR TKIs, whereas mutations in the ECD exhibit greater sensitivity to second-generation EGFR TKIs[4].

We next evaluated the sensitivity of EGFR R252C to four EGFR-TKIs (including afatinib, gefitinib, erlotinib, and osimertinib) by examining EGFR R252C-induced ERK1/2 activation and tumor cell proliferation. Immunoblotting analysis revealed that afatinib inhibited R252C-induced ERK1/2 phosphorylation in a dose-dependent manner, while gefitinib, erlotinib, and osimertinib exhibited minimal effects on phospho-ERK1/2 levels (Supplementary Fig. 5a, b). Supplementary Fig. 5c provides validation of the therapeutic efficacy of afatinib, gefitinib, erlotinib, and osimertinib. Consistent with the signaling data, afatinib exhibited stronger inhibition on R252C-driven proliferation than gefitinib and erlotinib, while osimertinib showed no anti-proliferative activity (Fig. 4a, b and Supplementary Fig. 6a). Collectively, these results suggested EGFR R252C mutation confers sensitivity to afatinib.

We then treated GSC387 cells stably expressing HA-EGFR WT or R252C with afatinib. Similarly, afatinib fully suppressed R252C-driven proliferation in GSC387 cells (Supplementary Fig. 6b). In addition, colony formation assays also demonstrated that U87, U251 and H1299 cells harbored EGFR R252C mutation had much less colonies formation after afatinib treatment than untreated cells (Fig. 4c). These results showed that afatinib can effectively suppressed the proliferative capacity of EGFR R252C-mutated cells.

To directly test the ability of afatinib to reduce tumor growth, we administered luciferase-expressing U87 cells with or without the EGFR R252C mutation, via intracranial injection into randomized athymic nude mice, and subsequently treated these mice with afatinib. Bioluminescence imaging of these mice showed that afatinib inhibited tumor growth in both EGFR WT and R252C tumor cells. Importantly, it showed a more pronounced inhibitory effect on tumor growth in cells with EGFR R252C (Fig. 4d). Meanwhile, we orthotopically injected luciferase-expressing H1299 cells with or without EGFR R252C mutation into the left lung of athymic nude mice and gave afatinib treatment. Similarly, the results indicated that afatinib more effectively inhibited the growth of lung tumors in mice with H1299 cells expressing EGFR R252C compared to EGFR WT cells, suggesting that afatinib can inhibit tumor growth induced by the EGFR R252C mutation in vivo. (Fig. 4e). Furthermore, in U87 engraftment mouse models, tumors expressing EGFR R252C exhibited significantly reduced overall survival compared to WT controls (median survival: 52 vs. 59 days; $p < 0.001$), aligning with their aggressive proliferative phenotype. Notably, afatinib treatment prolonged the survival of R252C-bearing mice. These findings were recapitulated in H1299 xenograft models, where EGFR R252C expression similarly correlated with worse survival outcomes (median survival: 39 vs. 49 days for WT; $p = 0.002$). Afatinib again demonstrated therapeutic efficacy, increasing the median survival of R252C-bearing mice (Supplementary Fig. 6c).

The patient with multifocal lung cancer and glioma, harboring the EGFR R252C mutation as previously described, showed disease progression despite multiple treatment regimens, including platinum-based chemotherapy (cisplatin), chemotherapy combined with bev-acizumab, and immunotherapy. Based on our prior experimental findings that afatinib inhibits the growth of EGFR R252C mutant tumor cells, we decided to administer afatinib therapy. Notably, following afatinib treatment, the patient exhibited remarkable clinical improvement, with CT and MRI scans revealing a marked reduction in lung and brain tumor burden (Fig. 4f). This clinical benefit persisted for approximately four years, ultimately leading to the patient's discharge (Fig. 4g). These findings suggest that afatinib is an effective therapeutic option for patients harboring this mutation and provide compelling

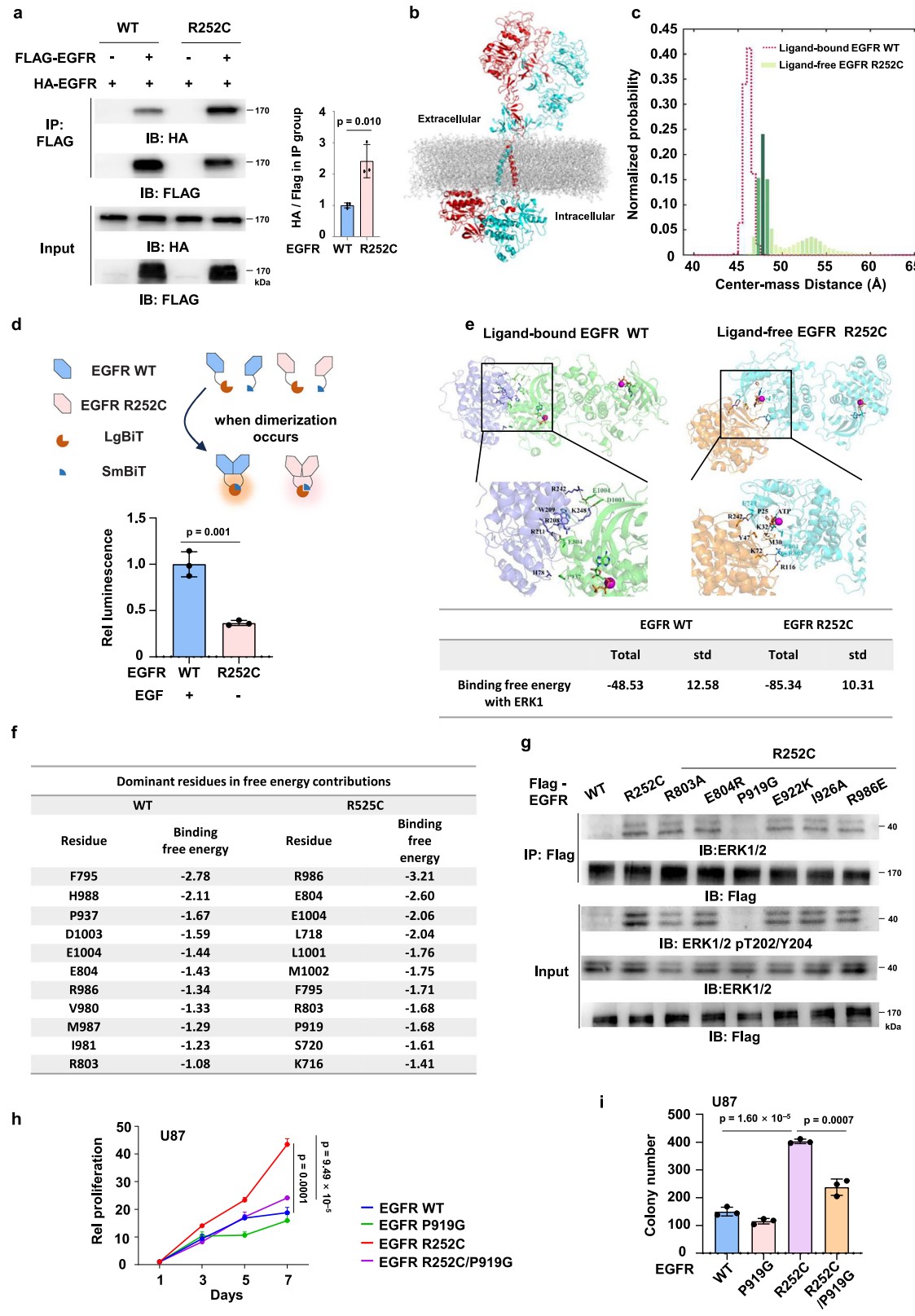

evidence supporting its clinical utility in the treatment of EGFR ECD mutations.

## Discussion

Functional studies on EGFR KD mutations have demonstrated that these mutants primarily activate the Akt and STAT signaling pathways, thereby promoting tumor cell survival[16]. In contrast, EGFR ECD mutations exhibit diverse functional mechanisms. For instance, EGFRVIII, a well-known EGFR ECD mutant, activates the PI3K-AKT pathway in tumor cell lines[17], whereas another representative ECD mutation, EGFR A289V, drives tumor cell invasion predominantly through the MAPK pathway[6]. These observations underscore the mechanistic complexity and heterogeneity of oncogenic signaling driven by ECD mutations in tumorigenesis and progression. However,

**Fig. 3 | Structural basis of EGFR R252C-dependent ERK1/2 activation.**
**a** 293T cells were co-transfected with EV, HA-Tagged or Flag-Tagged WT or R252C. Flag-EGFR was immunoprecipitated using anti-Flag agarose beads. Semi-quantitative scoring was carried out. The HA to Flag ratio in the IP samples was quantified and normalized to the EGFR WT group. **b** Conformations of the ligand-free EGFR R252C whole model, which contain the ECD, TM, JM and KD. **c** The distribution of the center-of-mass between the two kinase domains in the simulated ECD + TM + JM + KD dimers. The values of Center-mass Distance (Å) were colored in green and salmon for ligand-free EGFR R252C and ligand-bound EGFR WT, respectively. **d** Schematic diagram of the NanoBiT protein-protein interaction assay. EGFR WT or R252C-SmBiT together with EGFR WT or R252C-LgBiT and control Renilla plasmid were used to transfect HEK293T cells. Cells transfected with EGFR WT were serum-starved for 24 h before treated with 10 mg/ml EGF for 30 min. Biosensor activity was determined 36 h after transfection using the NanoBiT assay.

Relative luminescence was normalized to the EGFR WT group. **e** The binding mode of ERK1 and KD dimer in EGFR WT and R252C, and the total binding free energy were shown. In conformation, the KD dimer is colored in cyan and green in EGFR WT and R252C, respectively, and the ERK1 in orange and slate, respectively. While in the binding site model, the dominant residues shown in sticks and colored, respectively. **f** The dominant residues of the EGFR in the complex of EGFR and ERK1 binding are listed, and all energies are in units of kcal/mol. **g** Immunoprecipitation assay was performed with HEK293T cells stably expressing Flag-EGFR WT, Flag-EGFR R252C, and indicated mutants. **h** Cell proliferation assay was performed with U87 cells stably expressing Flag-EGFR WT or mutants. Relative cell proliferation of U87 cells was normalized to day 1 value. **i** Colony formation assay was performed with U87 cells stably expressing Flag-EGFR WT or mutants. **a, d, h, i.** Data represent the mean ± s.d. of three biologically independent experiments (unpaired, two-tailed t-test).

compared to the extensively studied KD mutations, the biological significance and downstream signaling consequences of ECD mutations remain poorly understood, highlighting a critical gap in current EGFR research. To investigate the mechanism behind EGFR R252C mutation, we examined three downstream pathways of EGFR. Contrary to previously reported activation ways of ECD mutations, we found that EGFR R252C does not phosphorylate EGFR as other ECD mutations do. Instead, it directly activates ERK1/2, leading to increased cell proliferation (Fig. 5).

Different EGFR mutations drive ligand-independent activation by inducing distinct conformational changes. EGFR KD mutations, such as exon 19 deletions and exon 20 insertions, relieve autoinhibition by affecting the conformational stability of the C-helix[18], while ECD mutations, such as EGFRVIII and R84K, promote activation by disrupting inhibitory interactions between ECDs[19,20]. These mechanisms highlight the diverse structural impacts of EGFR mutations, which will be further explored to identify novel therapeutic strategies. Collectively, these alterations maintain an open conformation of EGFR, rendering it more prone to active states even in the absence of ligand binding. In contrast, EGFR R252C, which does not induce reciprocal phosphorylation of EGFR monomers, can phosphorylate and activate ERK1/2 more easily in the absence of EGF. Of note, EGF ligand normally induces EGFR monomers to phosphorylate each other (auto-phosphorylation) for activation.

EGFR-TKIs, approved by the FDA as first-line treatments for NSCLC, have been confirmed to inhibit mutations in the KD of EGFR[21]. However, there are currently no approved therapeutic options for mutations in ECD[19]. In this study, we investigated the potential of EGFR inhibitor for the treatment of lung cancer and glioma patients harboring the EGFR R252C mutation, evaluating the efficacy of the second-generation EGFR TKI, afatinib, in animal models. Our findings demonstrated that afatinib treatment significantly reduced tumor burden and prolonged survival in mice bearing EGFR R252C tumors. Notably, motivated by our preliminary experimental results, we administered afatinib to a patient with multifocal lung cancer and glioma driven by EGFR R252C mutation. The treatment resulted in a substantial decrease in the patient's tumor volume in both the lung and brain, with no observed disease progression. Our research underscores that the relatively low rates of EGFR ECD mutations, as documented in existing databases, should not be disregarded, as they offer critical insights for disease assessment and treatment, which hold significant implications for patient outcomes. However, our study has a certain limitation: we have only collected data from one patient with an EGFR R252C mutation. More clinical data are needed to evaluate whether lung cancer and glioma patients with EGFR R252C could potentially benefit from afatinib treatment.

The limited efficacy of EGFR-targeted therapies in GBM was attributed to poor blood-brain barrier (BBB) permeability and diverse resistance mechanisms[22]. As highlighted by Reardon et al., afatinib demonstrated minimal single-agent activity in unselected recurrent GBM cohorts, and its combination with temozolomide failed to improve PFS-6 rates or median PFS[23]. However, our study revealed a distinct pharmacological profile of afatinib against the EGFR R252C mutation. Specifically, in vitro, afatinib exhibited superior inhibitory potency for EGFR R252C compared to other EGFR-TKIs (Supplementary Fig. 6a); in vivo, afatinib treatment significantly suppressed R252C-driven tumor growth and extended survival in xenograft models (Fig. 4d, e and Supplementary Fig. 6c). Importantly, afatinib effectively suppressed primary tumor growth and extended progression-free survival in a patient with multifocal lung cancer and glioma driven by EGFR R252C (Fig. 4f, g). These results could be attributed to the EGFR R252C mutation potentially altering BBB permeability, thereby creating a distinct therapeutic scenario that differs from conventional GBM paradigms. While the clinical implications of these observations require validation in larger patient cohorts, our data strongly support reconsideration of afatinib's therapeutic utility in molecularly defined GBM subtypes harboring this mutation. Of note, several case reports have documented prolonged responses to afatinib in glioblastoma patients[24,25], supporting further investigation of afatinib in GBM, particularly through biomarker-guided patient selection.

## Methods

### Ethics statement
Ethical approval for the human participant was approved by the Ethics Committee in Clinical Research (ECCR) of the First Affiliated Hospital of Wenzhou Medical University and the study was performed in accordance with the approved protocol. Informed consent was obtained from the patient's family for both research and publication of de-identified data included in this article. No compensation was provided to the participant.

All animal experimental protocols were approved by the Institutional Animal Care and Use Committee (IACUC) of the Shanghai Institute of Biochemistry and Cell Biology (approval number: SIBCB-S355-2312-40) and complied with all relevant ethical regulations.

### Mice
Six-week-old female BALB/c athymic nude mice were purchased from Lingchang Biotech (Shanghai, China). Littermates were randomly assigned to experimental groups. Sex was not considered as a biological variable in the study design. All experiments were conducted with female mice to limit complications of male territorial behavior and fighting during long-term cancer experiments. Mice were housed under specific pathogen-free conditions with a standard chow diet and maintained at a room temperature of 20–26 °C with 40–70% humidity on a 12-h light/12-h dark cycle. In accordance with the approved IACUC protocol, in vivo experiments were terminated at predefined time points or when mice reached humane endpoints, including a > 20% body weight loss or signs of severe distress (such as lethargy or impaired mobility). Mice were monitored daily, and euthanasia was performed according to IACUC-approved protocols once the endpoint

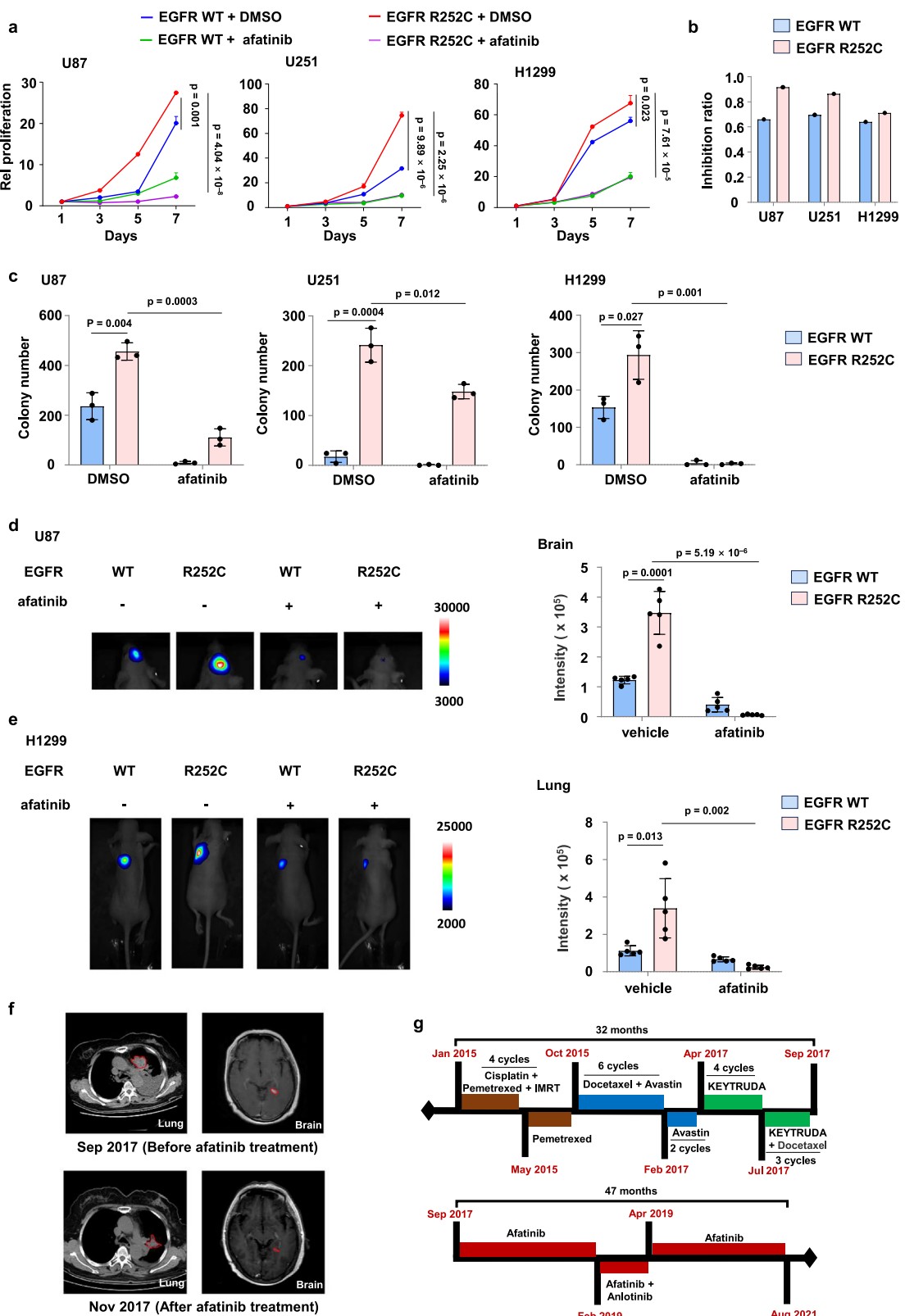

criteria were met. No mouse exceeded these ethical limits during the study.

## Antibodies

Rabbit monoclonal antibodies against HA (3724S), phospho-MEK1/2 (9154S), MEK1/2 (8727S), phospho-AKT (4056S), STAT3 (63585S), phospho-EGFR Y845 (6963S), phospho-EGFR Y1173 (4407S), EGFR (4267S) and rabbit polyclonal antibodies against phospho-EGFR Y992 (2235S), phospho-EGFR Y1045 (2237S), phospho-EGFR Y1086 (2220S), phospho-EGFR Y1148 (4404S), and phospho-EGFR Y1068 (2234S) were obtained from Cell Signaling Technology. Mouse monoclonal antibodies against GST (66001-2-Ig), His-tag (66005-1-Ig) and rabbit monoclonal antibody against Flag (20543-1-AP) were obtained from Proteintech Group. Mouse monoclonal antibody against phospho-

**Fig. 4 | Afatinib inhibits EGFR R252C-driven tumor cell proliferation and tumor growth in vivo. a**, **b** U87, U251, and H1299 cells with or without EGFR R252C mutation were treated with either vehicle (DMSO) or 1 μM afatinib. Cell proliferation assay (**a**, **b**) and colony formation assay (**c**) were performed in these cells. Relative cell proliferation of U87, U251, and H1299 cells was normalized to the day 1 value (**a**). The inhibition ratio of cell proliferation by afatinib was calculated at day 7 (**b**). **d** Luciferase-expressing U87 cells with or without EGFR R252C were intracranially injected into randomized athymic nude mice (five mice per group). After 35 days, afatinib was administered via oral gavage (10 mg/kg body weight) for a duration of 14 days. Bioluminescence imaging of mice was carried out to examine tumor growth. Data represent the mean ± s.d. of five mice. **e** Luciferase-expressing H1299 cells with or without EGFR R252C were injected into the left lung of randomized athymic nude mice (five mice per group). After 21 days, afatinib was administered via oral gavage (10 mg/kg body weight) for a duration of 14 days. Bioluminescence imaging of mice was carried out to examine tumor growth. Data represent the mean ± s.d. of five mice. **f** Chest CT and brain MRI images of a patient with multifocal lung cancer harboring EGFR R252C mutation, before and after treatment with afatinib. **g** Intact treatment timeline of the patient harboring EGFR R252C mutation. a,c Data represent the mean ± s.d. of three biologically independent experiments. **a**, **c**, **d**, **e**. Unpaired, two-tailed t-test.

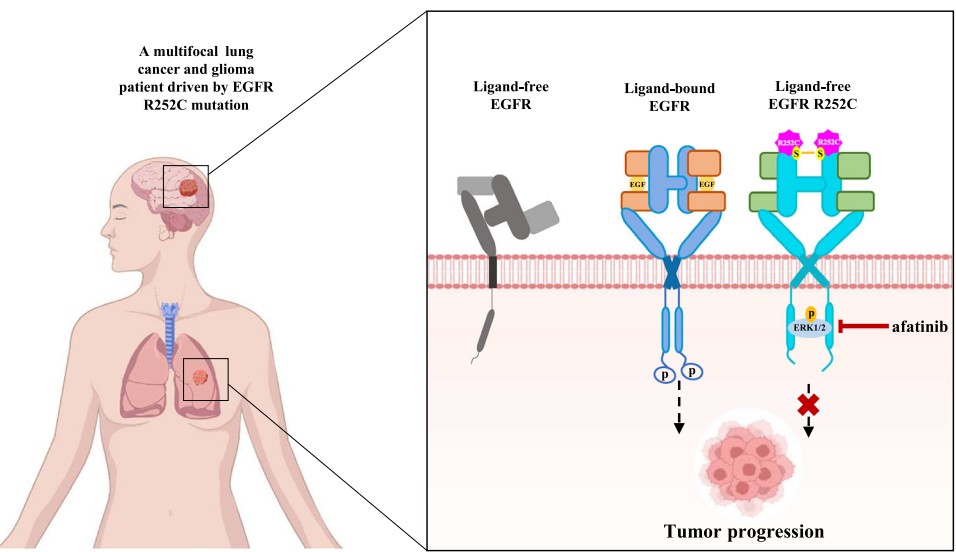

**Fig. 5 | Schematic model of an alternative EGFR activation by a patient-derived R252C mutation promotes cancer progression.** The R252C mutation promotes C252-C252 disulfide bond-mediated EGFR dimerization, inducing a conformational change of EGFR. This structural alteration enhances direct interaction between EGFR and ERK1/2, resulting in sustained ERK1/2 activation that drives tumor cell proliferation and tumor growth. In clinical application, afatinib effectively suppresses EGFR R252C-driven primary tumor progression and significantly extends progression-free survival in a patient with multifocal lung adenocarcinoma and glioma. Certain elements in schematic model were created with BioGDP.com[44].

ERK1/2 (sc-81492) and rabbit polyclonal antibody against ERK1/2 (sc-94) were purchased from Santa Cruz Biotechnology. Rabbit polyclonal antibodies against EGFR (A11351), AKT (A11016) and phospho-STAT3 (AP0070) were brought from Abclonal Technology. Mouse monoclonal antibody against Tubulin (T5201) was purchased from MilliporeSigma. The following secondary antibodies were used: goat-anti-mouse IgG second antibody (31160, Thermo); goat-anti-rabbit IgG second antibody (31210, Thermo). The primary antibodies were used at a 1:1000 dilution and the secondary antibodies were used at 1:3000 dilution for immunoblotting.

### Reagents

Puromycin (540411-100MGCN) was bought from Merck/Millipore (Darmstadt, Germany). DNA transfection reagent Hieff Trans Liposomal Transfection Reagent (H17520) was purchased from Yeasen Biotechnology (Shanghai, China). Flag peptide was obtained from Abclonal (Wuhan, China). ATP was purchased from InvivoGen (CA, USA). EGFR inhibitors (afatinib, gefitinib, erlotinib and osimertinib) and ERK1/2 inhibitor (ravoxertinib) was bought from MedChemExpress (New Jersey, USA). EGF and bFGF were purchased from Novoprotein (Suzhou, China). B27, sodium pyruvate and GlutaMAX were bought from Thermo Fisher Scientific.

### DNA constructs and mutagenesis

PCR-amplified human EGFR, ERK1, ERK2 were cloned into pCDH-Flag, pCDH-HA, pET28a-sumo, and pCold-GST vectors. Mutations were constructed by using the QuikChange site-directed mutagenesis kit (Stratagene, La Jolla, CA, USA).

The pGIPZ shNT was generated with the control oligonucleotide 5′-GCTTCTAACACCGGAGGTCTT-3′. pGIPZ human *MEK1* shRNA was generated with 5′-CTAGATGTTTAACAAATCT-3′ oligonucleotide. pGIPZ human *MEK2* shRNA was generated with 5′-CCGAGAGAA GCACCAGATC-3′ oligonucleotide.

### Immunoprecipitation and immunoblotting analysis

Extraction of proteins with a modified buffer from cultured cells was followed by immunoprecipitation and immunoblotting with corresponding antibodies, as described previously[26].

### Cell culture

HEK293T (GNHu17), H1299 (TCHu160), U87 (TCHu138), and U251 (TCHu 58) were obtained from the cell library of the Chinese Academy of Science and maintained in Dulbecco's Modified Eagle's Medium (DMEM) supplemented with 10% FBS and streptomycin–penicillin. Glioma stem cell line GSC387 was a gift from Prof. Huairui Yuan at CAS Center for Excellence in Molecular Cell Science. GSC387 cells were maintained in Neurobasal medium supplemented with B27, 20 ng/ml recombinant human EGF, 20 ng/ml recombinant human bFGF, sodium pyruvate, GlutaMAX and streptomycin-penicillin. Cells were incubated in 5% $CO_2$ at 37 °C. All cell lines were authenticated using the short tandem repeat method and tested negative for mycoplasma.

## Transfection

U87, U251, and H1299 cells were seeded in 35 mm, 60 mm, or 100 mm plates and transfected with the indicated plasmids using Hieff Trans Liposomal Transfection Reagent according to the manufacturer's instructions.

## Cell proliferation assay

$1 \times 10^3$ H1299, U87, or U251 cells in DMEM with 10% FBS were seeded per well in a 96-well plate. After culturing for 1, 3, 5, and 7 days, cells were collected. Then, cells were fixed with trichloroacetic acid and stained with SRB solution, followed by rinsing the plates four times with 1% acetic acid and solubilizing the protein-bound dye with 10 mM Tris base solution. The absorbance was measured at 510 nm.

## Generation of EGFR R252C knock-in cells

To establish EGFR R252C knock-in cells, the pX330-mCherry plasmid containing EGFR-sgRNA and the pCDH-Puro plasmid containing the donor fragment were co-transfected into U87, U251, and H1299 cells, respectively. 48 h after transfection, the cells were treated with 5 μg/ml puromycin for another 48 h. The cells were then sorted using a Sony MA900 sorter into a 96-well plate to isolate mCherry-positive single-cell clones. Two weeks later, genomic DNA was extracted from each clone, and successful knock-in was confirmed by PCR amplification followed by sequencing. The gRNA was designed based on the sequence: 5′-CCTTGCACGTGGCTTCGTCT-3′.

## In vitro kinase assay

In brief, the bacterially purified recombinant His-ERK1 or GST-ERK2 proteins were heated at 95 °C for 10 min. Flag-EGFR WT or R252C proteins were immunoprecipitated from 293T cells using anti-Flag agarose beads. Then 1 μg of His-ERK1 or GST-ERK2 proteins were incubated with Flag-EGFR WT or R252C proteins in kinase buffer (50 mM HEPES pH 7.5, 10 mM $MgCl_2$, 1 mM EGTA, 0.1 mM $Na_3VO_4$, and 1 mM DTT). 20 μM ATP was added to the kinase buffer to start the reaction. The reactions were performed in a total volume of 50 μl at 37 °C for 30 min and then terminated by adding SDS-PAGE loading buffer.

## Purification of recombinant proteins

His-ERK1 or GST-ERK2 was expressed in bacteria and purified. Briefly, the vectors expressing His-ERK1 or GST-ERK2 were used to transform BL21/DE3 bacteria. Then, 0.5 mM isopropyl-beta-D-thiogalactopyranoside (IPTG) was used to induce protein expression at 16 °C for 20 h. Cell pellets were collected and sonicated in PBS with the addition of proteasome inhibitors before centrifugation at $12,000 \times g$ for 30 min at 4 °C. Cleared lysates were then bound to Ni-NTA resin (GenScript) with rolling at 4 °C for 4 h. Beads were washed extensively before eluting for 1 h in elution buffer (PBS, pH 7.4, plus 500 mM imidazole). Eluted proteins were then dialyzed against PBS.

## Colony formation assay

For colony formation assay, 4000 cells per well were plated in a 6-well plate in triplicate and cultured for 14 days before staining viable colonies with crystal violet.

## Non-reducing SDS-PAGE

Non-reducing SDS-PAGE was performed to analyze protein samples under denaturing but non-reducing conditions. Cells were lysed in lysis buffer (50 mM Tris-HCl pH 7.5, 150 mM NaCl, 1% Triton X-100) supplemented with protease inhibitors. Protein extracts were mixed with 2× non-reducing loading buffer (50 mM Tris-HCl pH 6.8, 10% SDS, 10% glycerol, 0.1% bromophenol blue). Electrophoresis was carried out using 8% polyacrylamide gels with standard Bis-Tris running buffers. Following electrophoresis, proteins were transferred to PVDF membranes for subsequent immunoblotting analysis.

## Protein cross-linking analysis

Protein cross-linking analysis was performed by treating cells with 0.1% glutaraldehyde in PBS for 2 min at room temperature to stabilize protein-protein interactions, followed by quenching with 100 mM glycine for 5 min. Cells were then lysed in lysis buffer (50 mM Tris-HCl pH 7.5, 1% SDS) containing protease inhibitors. The cross-linked protein samples were mixed with 4× SDS loading buffer (100 mM Tris-HCl pH 6.8, 4% SDS, 20% glycerol, 1.4% β-mercaptoethanol, 0.1% bromophenol blue) and analyzed by 8% SDS-PAGE, followed by immunoblotting analysis.

## Intracranial injection

$2 \times 10^5$ (in 5 μl of DMEM per mouse) luciferase-expressing U87 cells with or without EGFR R252C mutation were injected into randomized 6-week-old female athymic nude mice by intracranial injection. After 35 days post-inoculation, mice were treated with afatinib for two weeks. Afatinib (10 mg/kg body weight) or saline was administered orally by gavage, 5 days per week. Five mice per group in each experiment were included. Bioluminescence imaging of mice injected with U87 cells was performed 49 days after implantation.

For the mouse survival analysis, the observation period was terminated upon the death of the last mouse in the vehicle-treated R252C group. At this endpoint, the surviving mice in the other group remained in good condition, with no significant weight loss or other adverse effects observed. These surviving mice were right-censored to uphold ethical standards and statistical integrity, as the Kaplan-Meier method appropriately handles censored data[27–29].

## Intrapulmonary injection

$2 \times 10^6$ (in 50 μl mixture of 25 μl of DMEM and 25 μl of Matrigel per mouse) luciferase-expressing H1299 cells with or without EGFR R252C mutation were injected into randomized 6-week-old female athymic nude mice by orthotopic pulmonary injection. After 21 days post-inoculation, mice were treated with afatinib for two weeks. Afatinib (10 mg/kg body weight) or saline was administered orally by gavage, 5 days per week. After 35 days of inoculation, bioluminescence imaging of these mice was performed.

For the mouse survival analysis, the observation period was terminated 10 days after the death of the last mouse in the vehicle-treated R252C group. At this endpoint, the surviving mice in other group remained in good condition, with no significant weight loss or other adverse effects observed.

## Bioluminescence imaging

Mice were anaesthetized with isoflurane inhalation and were subsequently intraperitoneally injected with 200 μl of 7.5 mg/ml D-luciferin. Bioluminescence imaging was initiated 10 min post-injection using a Tanon-5200 Chemiluminescent Imaging System (Tanon), with an exposure time of 5 min. All bioluminescent data were collected and analyzed using the Tanon-5200 Chemiluminescent Imaging System (version 1.0).

## NanoBiT assay

EGFR WT or R252C-SmBiT plasmid, EGFR WT or R252C-LgBiT plasmid, and control Renilla plasmid were used to transfect HEK293T cells in 12-well plates using Lipofectamine 2000 (Invitrogen) according to the manufacturer's protocol. After 36 h, the cells were harvested for the measurement of luciferase activities. The relative levels of luciferase activity of NanoLuc were normalized to the levels of luciferase activity of control Renilla.

## Molecular dynamics simulations

Atomistic molecular dynamic simulations of initial models were carried out in the AMBER18 program using AMBER14SB force field for protein[30,31], Lipid17 force field for lipid[32,33], and the parameters for the

nucleotides (ATP) were obtained from the parameters reported previously[34].

The simulations adopted the divide and conquer strategy, followed the strategy used by Arkhipov et al.[12] for constructed the active and inactive EGFR WT dimer model. The whole EGFR R252C model was divided into ECD, TM, JM, and KD, where the division was the same as in the active and inactive dimer construction[12], then the components were assembled into larger models. The disulfide bond covalently links the sulfur atoms between the close spatial proximity cysteine residues, and the C252 from each monomer formed the disulfide bond, which cross-linked the two monomers. The simulations were performed in following order: ECD, ECD + TM, ECD + TM + JM, ECD + TM + JM + KD models. And the model lipid consisted of 100% POPC lipids in extracellular leaflets, and 70% POPC and 30% negatively charged POPS lipids in intracellular leaflets, which the model was widely used in the membrane protein[35]. Based on the conformation of the EGFR R252C KD dimer, two initial molecular models were obtained: one of the ligand-free EGFR R252C KD dimer and ERK1, and the other of the ligand-bound EGFR WT KD dimer with ERK1, which was used as the control example. The models were generated from the results of ZDOCK[36].

The protein systems contain the dimer ECD (EGFR WT, EGFR R252C) system and dimer KD with ERK1 (EGFR R252C). Each protein system was neutralized with a number of sodium ions and then immersed in a solvent box filled with TIP3P water molecules[37], to warrant a distance of at least 10 Å between the surface of each protein model and the water box edge. The protein systems were subject to energy minimization in three stages to remove the bad contacts. Firstly, the solvent and the neutralized ions were minimized by holding the protein and ligand using a restraint with a strength of 100 kcal/(mol Å$^2$), and then the minimization was performed by holding the protein and ligand using a constraint of 10 kcal/(mol Å$^2$). Finally, the systems were minimized by removing any constraints. Each stage was performed using the steepest descent minimization of 1000 steps, followed by a 9000-step conjugate gradient minimization. NVT (constant Number of atoms, Volume and Temperature) simulations were carried out by heating the whole system linearly with time, gradually from 100 to 300 K in the first 300 ps, and the Berendsen thermostat[38] was used to maintain the temperature of the system. Subsequently, the system was equilibrated at a temperature of 300 K for 1 ns, followed by a NPT (constant Number of atoms, Pressure, and Temperature) production run. During the heating stage, all the protein and ligands were restrained by a restraint of 100 kcal/(mol Å$^2$), and under the equilibration stage, the restraint strength was decreased to 10 kcal/(mol Å$^2$). During the NPT production run, the Berendsen barostat[39] was used to control the pressure at 1 atm, and the Langevin thermostat was employed to control the temperature of the systems at 300 K. Five independent 6 μs (total 30 μs) production simulations of ligand-free R252C ECD were performed to obtain the stable conformation of the extracellular dimer.

The lipid systems contain ECD + TM, ECD + TM + JM, and ECD + TM + JM + KD. The semi-isotropic pressure coupling was employed using the Monte Carlo barostat, which controlled the pressure at 1 bar with a coupling constant of 5 ps when the production run was performed. To maintain the stability of the lipid system, all simulations were performed above the experimental liquid-crystalline phase transition temperature (~315 K for pure dipalmitoyllecithin)[40] with a time constant of 1 ps.

All bonds associated with hydrogen atoms were constrained by employing the SHAKE algorithm[41], and the Hydrogen Mass Repartitioning (HMR) method was adopted, such that the integration time step of 4 fs could be used. A cutoff value of 12 Å was set for nonbonded interactions, and the Particle Mesh Ewald method[42] was employed for treating electrostatic interactions. For each system, five independent molecular dynamic simulations were carried out using different velocities that were randomly generated at the beginning of the simulations and a long production run, totally up to 80 μs, only for ligand-free EGFR R252C model.

The last 500 ns of each simulation were used for analysis, and the analysis of distance, angle, and native contacts, etc., was performed by the cpptraj module in AMBER[31].

## Principal component analysis (PCA)

PCA provides orthogonal eigenvectors and eigenvalues according to covariance matrix diagonalization[30]. The elements Cij in the matrix are defined as:

$$C_{ij} = \frac{\left\langle \Delta r_i \times \Delta r_j \right\rangle}{\left( \left\langle r_i^2 \right\rangle \left\langle r_j^2 \right\rangle \right)^{1/2}} \tag{1}$$

where $\Delta r_i / \Delta r_j$ indicated the deviation of the Cα atom of the ith residue from its mean position. The value of Cij fluctuates between −1 and 1. A positive Cij value indicated the correlation motion between the ith residue and the jth residue, while a negative Cij value represented the inverse correlation motion. The eigenvalues indicated the magnitude of motions along a direction. In general, the first few principal components (PCs) described the most important slow system modes related to the functional motions of a biomolecular system.

## MM-GBSA calculation

To understand the interaction between the two dimers, the binding free energies were calculated using the MM-GBSA method. For each complex, 500 snapshots were extracted from the last 100 ns along the molecular dynamic trajectory at an interval of 200 ps. The MM-GBSA method[43] was performed to compute the binding free energies. The binding free energy ($\Delta G$) can be represented as:

$$\Delta G = \Delta E_{MM} + \Delta G_{sol} \tag{2}$$

where $\Delta E_{MM}$ is the difference of molecular mechanic energy between the complex and each binding partner in the gas phase, $\Delta G_{sol}$ is the solvation free energy contribution to binding and $T\Delta S$. $\Delta E_{MM}$ is further divided into two parts:

$$\Delta E_{MM} = \Delta E_{ele} + \Delta E_{vdW} \tag{3}$$

where $\Delta E_{ele}$ and $\Delta E_{vdW}$ are described as the electrostatic interaction and van der Waals energy in the gas phase, respectively. The solvation free energy is expressed as:

$$\Delta G_{sol} = \Delta G_{gb} + \Delta G_{np} \tag{4}$$

where $\Delta G_{gb}$ and $\Delta G_{np}$ are the polar and non-polar contributions to the solvation free energy, respectively.

## Statistics and reproducibility

All the results were expressed as the mean ± s.d. No statistical method was used to predetermine sample size. The exact sample size (n) for each experimental group was indicated in the figure legends. No data were excluded from the analyses. All statistical analyses were performed using GraphPad Prism 8. For comparisons between two unpaired independent groups, the two-tailed Student's t test was used. Statistical significance was defined as $p < 0.05$. Randomization was applied to the allocation of experimental groups to minimize selection bias. Throughout the experiments and data analysis, investigators were blinded to group assignments to prevent any potential bias during outcome assessment.

**Reporting summary**

Further information on research design is available in the Nature Portfolio Reporting Summary linked to this article.

## Data availability

Source data are provided with this paper.

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

## Acknowledgements

This work was supported by National Key R&D Program of China (2022YFA0806200 and 2024YFA1306000) to W.Y.; the National Natural Science Foundation of China (32521007, 92357301, and 32025013) to W.Y.; the Strategic Priority Research Program of the Chinese Academy of Science (XDB0990000) to W.Y.; Science and Technology Commission of Shanghai Municipality (24J12800600) to W.Y.; Shanghai Municipal Science and Technology Major Project; CAS Project for Young Scientists in Basic Research (YSBR-014) to W.Y.; the Research Funds of Hangzhou Institute for Advanced Study, UCAS (2025HIAS-ZL014) to W.Y.; the foundation of Shanghai Key Laboratory of Thoracic Tumor Biotherapy (No. 2025SZ1701) to W.Y.; Shanghai Academy of Natural Sciences (SANS) to W.Y.; the National Natural Science Foundation of China (32470821) to Y. Zhang; the Shanghai Oriental Talent Program to Y. Zhang; the Shanghai Natural Science Foundation (23ZR1470100) to Y. Zhang; National Key R&D Program of China (2024YFA1306000) to Y. Zhang; Bethune Charitable Foundation (BCF-QYWL-ZL-2025-10) to W.L.; Wu Jieping Medical Foundation (320.6750.2024-16-2) to W.L.; National Key R&D Program of China (2023YFF1204903) to H.C.; the LiaoNing Revitalization Talents Program (No. XLYC2402023) to G.L.; the National Natural Science Foundation of China (82571814) to C.C. We gratefully thank Prof. Huairui Yuan at CAS Center for Excellence in Molecular Cell Science for providing GSC387 cell line. We also thank all the core facilities of CAS Center for Excellence in Molecular Cell Science for technical support.

## Author contributions

W.Y. conceived the study. W.Y. and Y. Zhang designed the study. Y. Zhang and Q.F. performed the experiments and data analysis. S.W., T.R., and X.W. provided experimental assistance. H.C. and G.L. designed the molecular dynamics strategy. H.C. and Y.L. performed the simulations and data analysis. C.C. and W.L. provided pathological data. H.G., D.G. and Y. Zhao assisted in reviewing the manuscript. W.Y., Y. Zhang, and Q.F. wrote the manuscript with comments from all authors.

## Competing interests

The authors declare no competing interests.
