## [Transparent Peer Review file · Nature Communications]

An alternative EGFR activation by patient-derived R252C mutation promotes cancer progression

Corresponding Author: Dr Weiwei Yang

Version 0:

Reviewer comments:

Reviewer #1

(Remarks to the Author)

Zhang et al. present a compelling study describing the identification and mechanistic characterization of a novel EGFR extracellular domain mutation, R252C, found in a patient with multifocal lung cancer and glioma. Using CRISPR-engineered isogenic models and orthotopic xenografts, the authors demonstrate that R252C enhances tumor proliferation via a non-canonical mechanism that bypasses typical EGFR autophosphorylation and upstream MAPK activation. Instead, R252C stabilizes an atypical EGFR dimer that directly phosphorylates ERK1/2, a mechanism supported by structural modeling and biochemical assays. The work is strengthened by the use of multiple model systems, elegant integration of computational and experimental data, and clinically relevant findings, including durable therapeutic response to afatinib in the index patient.

However, several limitations temper the overall conclusions. Key mechanistic claims—such as direct MEK-independent ERK activation and disulfide-stabilized dimerization—lack definitive experimental validation. Comparisons to other EGFR TKIs are missing, limiting interpretation of afatinib's selectivity. Quantitative analyses, especially for immunoblots and NanoBiT assays, are inconsistently applied, and survival data for xenograft models are absent. Additionally, concerns about CNS drug penetration and lack of target engagement data in GBM models raise questions about translational applicability.

Overall, this study offers a novel and potentially paradigm-shifting insight into EGFR biology, with clear therapeutic relevance, but would benefit from additional mechanistic and pharmacologic validation to fully support its conclusions.

Specific Comments:

1) Mechanistic validation of ERK activation: A major claim is that EGFR mediated ERK activation is direct and therefore bypasses canonical RAS-RAF-MEK signaling. However, this claim needs experimental validation. To improve on the mechanistic claims relating to direct EGFR mediated activation of ERK, the authors should test whether MEK/RAF/RAS inhibitors (e.g., trametinib, dabrafenib, tipifarnib) impact R252C-driven ERK1/2 phosphorylation as a way to directly assess MEK independence. The authors should also include genetic perturbation experiments (e.g., siRNA or CRISPR knockdown of MEK1/2) to further confirm bypass of canonical MAPK signaling.

2) Broaden analysis of EGFR autophosphorylation: To determine whether activation patterns of EGFR are site specific to Y1068, the authors should assess a broader panel of EGFR phosphorylation sites (e.g., Y845, Y992, Y1045, Y1086, Y1148, Y1173), all of which can differentially impact binding of adaptor/effector proteins. This analysis should be done +/- EGF stimulation to assess dynamic regulation.

3) Direct evidence for disulfide dimerization. While the molecular dynamics simulations offer a compelling mechanistic hypothesis in which R252C stabilizes a unique ECD dimer, more evidence is needed to support this conclusion. The authors should consider non-reducing PAGE or quantitative mass-spectrometry to directly demonstrate C252-C252 disulfide formation in cells through demonstration of an intact dimer under non-liganded conditions (PAGE) and/or to directly identify the relevant C252-C252 dimer peptide via MS. Moreover, these data could be complemented by a loss-of-function variant (e.g., C225S mutant), which presumably would lose disulfide formation capability, dimer stability (via NanoBiT or crosslinking assays), and fail to phosphorylate ERK1/2.

4) Confirmation of ERK docking site on EGFR. The authors claim the EGFR P919 residue is responsible for binding ERK1/2 in the R252C mutant. To support this finding, the authors could generate a single P919G mutant in the WT background to compare to the double R252C/P919G mutant.

5) Evaluation of sensitivity against other EGFR TKIs. Mutations in ECD vs KD can impact EGFR TKI sensitivity (Vivanco et al., Cancer Discovery). The authors should compare afatinib with other clinically relevant EGFR TKIs (gefitinib, erlotinib, osimertinib) in dose-response experiments using proliferation and signaling readouts (e.g., pERK1/2 levels). Importantly, they should include side-by-side comparison of TKI efficacy in WT vs. R252C cell lines to better define mutation-specific sensitivity.

6) Lack of Relevant Models: While the U87MG cell line can be beneficial for biochemical investigations, it does not represent GBM in a substantive manner. The authors should put in considerable effort to include more clinically relevant models (such as patient-derived gliomaspheres) to validate their findings. If they are unable to locate such models, the authors should clearly state this as a limitation of this study.

Minor points

- 1) Mouse survival should be included as an endpoint
- 2) Afatinib has extremely low CNS penetration and has failed in the clinic (Reardon et al. Neuro-Oncology. 2014). The authors should either provide direct evidence that these mutations are different from those previously targeted with afatinib in GBM or substantially state this as a limitation of their study, and to propose other drugs (see point 5) that have greater CNS exposures but can target this mutation effectively relative to WT EGFR
- 3) Improvement to data analysis and interpretation: A. Quantify immunoblots and NanoBiT assay results using appropriate densitometry or luminescence normalization, and include statistical analyses. B) Add expression/loading controls (e.g., total protein or luciferase levels) to validate NanoBiT assay comparisons. C) Clarify discrepancies in afatinib efficacy between WT and R252C cells (e.g., Fig. 4A) with fold-change metrics or normalized growth curves. D) Raw uncropped blots should be provided for review.

Reviewer #2

(Remarks to the Author)

Reviewer #3

(Remarks to the Author)

This manuscript describes a novel EGFR mutation in a patient with lung cancer and glioma and characterises the molecular mechanisms of EGFR signalling exerted by this EGFR mutant. The R252C mutation leads to enhanced dimer formation of the receptor, direct phosphorylation of ERK1/2 and enhanced proliferative responses in cell lines and xenograft tumor growth. The effects can be reversed by afatinib, suggesting a potential therapeutic strategy.

The paper is well written, presents a coherent story and uses state of the art technologies. In addition to the experimental work, there are structural molecular simulations to support the hypothesis and key validation experiments to substantiate the claims.

I have some minor comments to improve the quality of the manuscript:

1. The abstract states that the R252C mutation results in reduced auto-phosphorylation, but this is not substantiated by the results.
2. I would have liked to see more references in the introduction. Many statements are not backed up by the correct references.
3. Apart from the R252C mutation: were there other mutations in other genes observed? Was the status of other key oncogenes tested?
4. There are some minor typos throughout the text. Please run a spell checker over the manuscript.
5. Figures 2a and 2b need quantification from three independent experiments.
6. Is the Grb2-SOS-Kras axis not required for phosphorylation of ERKs? I wonder if a quick siRNA or inhibitor experiment could be done to verify that the direct binding of ERKs is sufficient for signalling by the R252C mutant.
7. Figure 3a: a negative control is missing.

Version 1:

Reviewer comments:

Reviewer #1

(Remarks to the Author)

Zhang et al. have addressed the major concerns from our first review and have added substantial new data that strengthen the central claim that EGFR R252C directly drives oncogenic ERK activation and is therapeutically targetable by afatinib. The revised manuscript and reviewer response detail our requested mechanistic experiments (disulfide-dependent dimerization, NanoBiT/crosslinking, P919 mutational tests), RAS/RAF/MEK inhibitor and genetic MEK depletion controls, and side-by-side TKI comparisons and in vivo survival analyses. Overall, the revisions are convincing and the authors should be commended for their effort and rigor that went into addressing the comments. However, a few revisions/additions are requested to further strengthen and clarify their conclusions

1. Major: Provide final updated survival data and Kaplan-Meier analyses with the publication. The survival endpoints of afatinib treatment strengthen the authors' translation claim. If data remains censored, please provide censoring details and an explanation of why full curves could not be presented.

2. Minor: Clarify why afatinib shows apparent mutation-selective efficacy and fails to target wild-type EGFR. Despite its known biochemical potency against wild-type EGFR (~0.5 nM; Li et al., Oncogene 2008, in Supplemental Fig. 2a, afatinib does not show a significant impact on pEGFR^{WT} and in Supplemental Fig. 5a neither afatinib nor other wild-type targeting EGFR TKIs (erlotinib, gefitinib) significantly suppress ERK phosphorylation, even at high concentrations (>1 μ M).

Reviewer #2

(Remarks to the Author)

Reviewer #3

(Remarks to the Author)

I would like to thank the authors for the additional data that has been provided and that addresses some of the concerns that were raised. However, there are still some aspects that require further clarification.

1. I find the legends in the supplementary figures too short to be able to understand what was done. This makes it very difficult to understand the figures. For example, in Supplementary figure 2: has this been done in full media with FBS, have the cells been stimulated, etc. In the new supplementary figure 2, it is thus not obvious to understand whether the inhibitors actually work. One would need to stimulate the cells with EGF in order to see whether the Ras, Raf and MEK inhibitor actually work at the concentration used (what is the concentration anyway? – not mentioned in the legend).
2. I still do not see a reduced auto-phosphorylation for R252C. Supplementary Figure 4e does not show this. In unstimulated cells, there is no phosphorylation visible in both wt and R252C cells. Also, R252C should be shown with EGF stimulation. One cannot compare unstimulated R252C to stimulated wt cells.

Version 2:

Reviewer comments:

Reviewer #3

(Remarks to the Author)

Well, I still have an issue with the auto-phosphorylation of the EGFR, but maybe this is just a misunderstanding. There is no auto-phosphorylation by wt nor the mutant receptor (i.e. phosphorylation without EGF stimulation, see supplementary figure 4e). But maybe what the authors mean to say is ABSENT auto-phosphorylation?

Response to the referees' comments:

Reviewer #1 (Remarks to the Author):

Zhang et al. present a compelling study describing the identification and mechanistic characterization of a novel EGFR extracellular domain mutation, R252C, found in a patient with multifocal lung cancer and glioma. Using CRISPR-engineered isogenic models and orthotopic xenografts, the authors demonstrate that R252C enhances tumor proliferation via a non-canonical mechanism that bypasses typical EGFR autophosphorylation and upstream MAPK activation. Instead, R252C stabilizes an atypical EGFR dimer that directly phosphorylates ERK1/2, a mechanism supported by structural modeling and biochemical assays. The work is strengthened by the use of multiple model systems, elegant integration of computational and experimental data, and clinically relevant findings, including durable therapeutic response to afatinib in the index patient.

However, several limitations temper the overall conclusions. Key mechanistic claims—such as direct MEK-independent ERK activation and disulfide-stabilized dimerization—lack definitive experimental validation. Comparisons to other EGFR TKIs are missing, limiting interpretation of afatinib's selectivity. Quantitative analyses, especially for immunoblots and NanoBiT assays, are inconsistently applied, and survival data for xenograft models are absent. Additionally, concerns about CNS drug penetration and lack of target engagement data in GBM models raise questions about translational applicability.

Overall, this study offers a novel and potentially paradigm-shifting insight into EGFR biology, with clear therapeutic relevance, but would benefit from additional mechanistic and pharmacologic validation to fully support its conclusions.

Response: Thank you for the constructive comments and insightful suggestions. To address the concerns raised, we have conducted additional experiments and expanded the discussion in the Discussion section. All changes we have made in the manuscript have been highlighted in yellow.

Specific Comments:

1) Mechanistic validation of ERK activation: A major claim is that EGFR mediated ERK activation is direct and therefore bypasses canonical RAS-RAF-MEK signaling. However, this claim needs experimental validation. To improve on the mechanistic claims relating to direct EGFR mediated activation of ERK, the authors should test whether MEK/RAF/RAS inhibitors (e.g., trametinib, dabrafenib, tipifarnib) impact R252C-driven ERK1/2 phosphorylation as a way to directly assess MEK independence. The authors should also include genetic perturbation experiments (e.g., siRNA or CRISPR knockdown of MEK1/2) to further confirm bypass of canonical MAPK signaling.

Response: Thank you for the constructive comment and suggestion. To investigate whether EGFR R252C-induced ERK1/2 activation is dependent on the canonical RAS/RAF/MEK pathway, we treated U251 cells harboring either EGFR WT or EGFR R252C with RAS inhibitor (tipifarnib), RAF inhibitor (dabrafenib), or MEK inhibitor (trametinib), respectively. As indicated by the result of Supplementary Fig. 2b-d, inhibition of RAS, RAF, or MEK failed

to block EGFR R252C-induced ERK1/2 activation, indicating that EGFR R252C-induced ERK1/2 activation is independent of the canonical RAS/RAF/MEK pathway. This conclusion was further validated by MEK1/2 depletion experiments in both U251 and H1299 cell lines (Supplementary Fig. 2e,f). The results were described in line 154.

2) Broaden analysis of EGFR autophosphorylation: To determine whether activation patterns of EGFR are site specific to Y1068, the authors should assess a broader panel of EGFR phosphorylation sites (e.g., Y845, Y992, Y1045, Y1086, Y1148, Y1173), all of which can differentially impact binding of adaptor/effector proteins. This analysis should be done +/- EGF stimulation to assess dynamic regulation.

Response: Thank you for the constructive comment and suggestion. To determine whether R252C-induced activation patterns of EGFR are site-specific to Y1068, we harvested U251 cells harboring either EGFR WT or EGFR R252C and U251 cells harboring EGFR WT treated with EGF (10 ng/ml). Immunoblotting analysis revealed that both EGFR WT and EGFR R252C cells exhibited comparable basal phosphorylation levels at Y845, Y992, Y1045, Y1086, Y1148, and Y1173 residues. In contrast, EGF-treated U251 cells exhibited increased phosphorylation levels at Y845, Y992, Y1045, Y1086, Y1148, and Y1173 residues (Supplementary Fig. 4e). The above results suggest that R252C-induced activation patterns of EGFR are not site-specific to Y1068. The results were described in line 243.

3) Direct evidence for disulfide dimerization. While the molecular dynamics simulations offer a compelling mechanistic hypothesis in which R252C stabilizes a unique ECD dimer, more evidence is needed to support this conclusion. The authors should consider non-reducing PAGE or quantitative mass-spectrometry to directly demonstrate C252-C252 disulfide formation in cells through demonstration of an intact dimer under non-liganded conditions (PAGE) and/or to directly identify the relevant C252-C252 dimer peptide via MS. Moreover, these data could be complemented by a loss-of-function variant (e.g., C252S mutant), which presumably would lose disulfide formation capability, dimer stability (via NanoBiT or crosslinking assays), and fail to phosphorylate ERK1/2.

Response: Thank you for the insightful suggestion. To determine whether R252C mutation stabilizes a unique ECD dimer of EGFR and whether the dimer formation is dependent on the C252-C252 disulfide, we transfected H1299 cells with HA-tagged EGFR WT, R252C, or R252S. The cells were harvested and analyzed by non-reducing SDS-PAGE to preserve disulfide bonds. As shown in Supplementary Fig. 3b, R252C mutation indeed promoted the dimer formation of EGFR, while R252S mutation failed to do so. The result confirmed that R252C stabilizes a unique EGFR ECD dimer specifically through C252-C252 disulfide bond formation. This conclusion was further validated by glutaraldehyde-mediated crosslinking experiments (Supplementary Fig. 3c). Besides, we transfected H1299 cells with HA-tagged EGFR WT, R252C, or R252S. The cells were harvested and analyzed by SDS-PAGE. As shown in Supplementary Fig. 3d, compared to EGFR WT, R252C mutation enhanced ERK1/2 activation, while R252S mutation failed to do so. These results confirmed that R252C mutation promotes the dimerization of EGFR to activate ERK1/2, which is dependent on C252-C252 disulfide bond. The results were described in line 187.

4) Confirmation of ERK docking site on EGFR. The authors claim the EGFR P919 residue is

responsible for binding ERK1/2 in the R252C mutant. To support this finding, the authors could generate a single P919G mutant in the WT background to compare to the double R252C/P919G mutant.

Response: Thank you for the constructive comment and suggestion. To confirm the role of P919 in mediating EGFR R252C-induced ERK1/2 activation and cell proliferation, we introduced the P919G mutation into both EGFR WT and R252C backgrounds. Co-immunoprecipitation assay demonstrated that the P919G mutation significantly abrogated the enhanced interaction between EGFR R252C and ERK1/2, concurrently suppressing R252C-driven ERK1/2 phosphorylation (Supplementary Fig. 4h). Notably, the P919G mutation had no discernible effect on either the EGFR-ERK1/2 interaction or basal ERK1/2 phosphorylation in the wild-type EGFR background (Supplementary Fig. 4h). Consistently, both cell proliferation and soft agar colony formation assays confirmed that the P919G mutation significantly attenuated EGFR R252C-driven proliferation (Fig. 3h,i and Supplementary Fig. 4i). The results were described in line 256.

5) Evaluation of sensitivity against other EGFR TKIs. Mutations in ECD vs KD can impact EGFR TKI sensitivity (Vivanco et. al., Cancer Discovery). The authors should compare afatinib with other clinically relevant EGFR TKIs (gefitinib, erlotinib, osimertinib) in dose-response experiments using proliferation and signaling readouts (e.g., pERK1/2 levels). Importantly, they should include side-by-side comparison of TKI efficacy in WT vs. R252C cell lines to better define mutation-specific sensitivity.

Response: Thank you for the constructive comment and suggestion. To evaluate the sensitivity of EGFR R252C to EGFR-TKIs, U251 and H1299 cells expressing EGFR WT or R252C were treated with increasing concentrations (0, 1, 5, 10 μ M) of four EGFR inhibitors: afatinib, gefitinib, erlotinib, or osimertinib. Immunoblotting analysis revealed that afatinib inhibited ERK1/2 phosphorylation in a dose-dependent manner, while gefitinib, erlotinib, and osimertinib exhibited minimal effects on phospho-ERK1/2 levels (Supplementary Fig. 5a,b). Supplementary Fig. 5c provides validation of the inhibitory efficacy of gefitinib, erlotinib, and osimertinib. Consistent with the signaling data, afatinib exhibited stronger inhibition on R252C-driven proliferation than gefitinib and erlotinib, while osimertinib showed no anti-proliferative activity (Supplementary Fig. 6a). Collectively, these results suggested that EGFR R252C mutation confers sensitivity to afatinib. The results were described in line 282.

6) Lack of Relevant Models: While the U87MG cell line can be beneficial for biochemical investigations, it does not represent GBM in a substantive manner. The authors should put in considerable effort to include more clinically relevant models (such as patient-derived gliomaspheres) to validate their findings. If they are unable to locate such models, the authors should clearly state this as a limitation of this study.

Response: Thank you for the constructive comment. To validate our findings in a clinically relevant GBM model, we utilized the patient-derived GSC387 glioma stem cell line, which maintains key molecular and phenotypic characteristics of primary glioblastoma. Using lentiviral-mediated gene delivery, we established stable GSC387 cell lines expressing either HA-tagged EGFR WT or the R252C mutant. As demonstrated in Supplementary Fig. 1a,b, expression of EGFR R252C significantly enhanced GSC387 proliferation compared to EGFR

WT. Mechanistically, immunoblot analysis revealed that the R252C mutation markedly increased ERK1/2 phosphorylation. Importantly, this mutant-enhanced ERK1/2 activation was effectively suppressed by afatinib treatment (Supplementary Fig. 2a). Consistent with these molecular observations, the proliferation advantage conferred by EGFR R252C was completely abolished upon afatinib exposure (Supplementary Fig. 6b). These results collectively demonstrate that the EGFR R252C mutation drives GSC proliferation through enhanced ERK1/2 activation, and that this oncogenic phenotype is therapeutically targetable by afatinib. The results were described in line 115, line 142 and line 292.

Minor points

1) Mouse survival should be included as an endpoint.

Response: We appreciate this insightful suggestion. In our revised study, we have incorporated mouse survival as a critical endpoint to complement the tumor growth and molecular analyses. Specifically, in U87 engraftment mouse models, log-rank analysis revealed that tumors expressing EGFR R252C exhibited significantly reduced overall survival compared to WT controls (median survival: 52 vs. 59 days; $p < 0.001$), aligning with their aggressive proliferative phenotype. Notably, afatinib treatment prolonged the survival of R252C-bearing mice. These findings were recapitulated in H1299 xenograft models, where EGFR R252C expression similarly resulted in worse survival outcomes (median survival: 39 vs. 49 days for WT; $p = 0.002$). Afatinib again demonstrated therapeutic efficacy, increasing the median survival of R252C-bearing mice (Supplementary Fig. 6c). The results were described in line 309.

2) Afatinib has extremely low CNS penetration and has failed in the clinic (Reardon et al. Neuro-Oncology. 2014). The authors should either provide direct evidence that these mutations are different from those previously targeted with afatinib in GBM or substantially state this as a limitation of their study, and to propose other drugs (see point 5) that have greater CNS exposures but can target this mutation effectively relative to WT EGFR.

Response: We appreciate the reviewer's insightful comment regarding afatinib's CNS penetration and clinical performance in GBM. Indeed, the limited efficacy of EGFR-targeted therapies in GBM was attributed to poor blood-brain barrier (BBB) permeability and diverse resistance mechanisms (PMID: 37805108). As reported by Reardon et al., afatinib demonstrated minimal single-agent activity in **unselected recurrent GBM cohorts**, and its combination with temozolomide failed to improve PFS-6 rates or median PFS (PMID: 25140039). However, our study revealed a distinct pharmacological profile of afatinib against the EGFR R252C mutation. Specifically, in vitro, afatinib exhibited superior inhibitory potency and selectivity for EGFR R252C compared to other EGFR-TKIs (Supplementary Fig. 6a); in vivo, afatinib treatment significantly suppressed R252C-driven tumor growth and extended survival in xenograft models (Fig. 4d,e, and Supplementary Fig. 6c). Importantly, afatinib effectively suppressed tumor growth and extended progression-free survival in a patient with multifocal lung cancer and glioma driven by EGFR R252C (Fig. 4f,g). These results could be attributed to the EGFR R252C mutation potentially altering BBB permeability or increasing afatinib-EGFR binding affinity, thereby creating a distinct therapeutic scenario that differs from conventional GBM paradigms. While the clinical implications of these observations require

validation in larger patient cohorts, our data support reconsideration of afatinib's therapeutic utility in molecularly defined GBM subtypes harboring this mutation.

Of note, several case reports have also documented prolonged responses to afatinib in glioblastoma patients (PMID: 26423602; PMID: 34787778), suggesting further investigation of afatinib in GBM, particularly through biomarker-guided patient selection. We have added this discussion to the Discussion section (line 375).

3) Improvement to data analysis and interpretation: A. Quantify immunoblots and NanoBiT assay results using appropriate densitometry or luminescence normalization, and include statistical analyses. B) Add expression/loading controls (e.g., total protein or luciferase levels) to validate NanoBiT assay comparisons. C) Clarify discrepancies in afatinib efficacy between WT and R252C cells (e.g., Fig. 4A) with fold-change metrics or normalized growth curves. D) Raw uncropped blots should be provided for review.

Response: We appreciate the constructive comments. In the revised manuscript, we have made revisions based on the suggestions of the reviewers. Specifically: (A) For immunoblots, we have performed densitometric analysis using ImageJ, normalized all signals to their corresponding loading controls, and added statistical comparisons (unpaired t-tests, n=3 biological replicates) in revised figures. For NanoBiT assays, we have added Renilla luciferase co-transfection as an internal control and normalized luminescence values to Renilla luciferase activity. Statistical comparisons (unpaired t-tests, n=3 biological replicates) were also performed. (B) We have added expression controls to validate NanoBiT assay comparisons (Supplementary Fig. 4c,d). (C) Regarding afatinib efficacy comparisons, we have now quantified the growth curves in Fig. 4a using normalized growth rates (relative to DMSO controls) and calculated fold-change differences in EGFR WT or R252C group at day 7 (Fig. 4b). (D) For blot transparency, all uncropped western blots with molecular weight markers have been compiled in the Source data.

Reviewer #3 (Remarks to the Author):

This manuscript describes a novel EGFR mutation in a patient with lung cancer and glioma and characterises the molecular mechanisms of EGFR signalling exerted by this EGFR mutant. The R252C mutation leads to enhanced dimer formation of the receptor, direct phosphorylation of ERK1/2 and enhanced proliferative responses in cell lines and xenograft tumor growth. The effects can be reversed by afatinib, suggesting a potential therapeutic strategy.

The paper is well written, presents a coherent story and uses state of the art technologies. In addition to the experimental work, there are structural molecular simulations to support the hypothesis and key validation experiments to substantiate the claims.

Response: We thank the reviewer for the positive comments and insightful suggestions. We performed additional experiments to address the concerns of the reviewer. All changes we have made in the manuscript have been highlighted in yellow.

Major points:

I have some minor comments to improve the quality of the manuscript:

1. The abstract states that the R252C mutation results in reduced auto-phosphorylation, but this is not substantiated by the results.

Response: Thank you for the constructive comment. We have included the results in the revised manuscript. To examine auto-phosphorylation levels of EGFR, we harvested U251 cells harboring either EGFR WT or EGFR R252C and U251 cells harboring EGFR WT treated with EGF (10 ng/ml). Immunoblotting analysis revealed that both EGFR WT and R252C cells exhibited comparable basal phosphorylation levels at Y845, Y992, Y1045, Y1086, Y1148, and Y1173 residues. In contrast, EGF-treated U251 cells exhibited increased phosphorylation levels at Y845, Y992, Y1045, Y1086, Y1148, and Y1173 residues (Supplementary Fig. 4e), demonstrating the reduced auto-phosphorylation of EGFR R252C, compared to the ligand-bound EGFR WT. The results were described in line 243.

Notably, in Fig. 3, we found that the simulated EGFR R252C model contains an active ligand-free extracellular dimer conformation, similar to the ligand-bound active EGFR WT extracellular dimer. However, the ligand-free EGFR R252C exhibits an increased distance between the KDs and an altered domain orientation. These changes are the primary reasons for the reduced auto-phosphorylation levels observed in EGFR R252C compared to ligand-bound EGFR WT.

2. I would have liked to see more references in the introduction. Many statements are not backed up by the correct references.

Response: We appreciate this helpful suggestion and have now expanded the introduction with additional references to better support our key statements.

3. Apart from the R252C mutation: were there other mutations in other genes observed? Was the status of other key oncogenes tested?

Response: Thank you for the comment. Comprehensive mutation profiling of key oncogenic

loci, including *NRAS* Q61/G13/G12, *PIK3CA* H1047, *EGFR* G719/L858, *BRAF* V600, and *KRAS* Q61/G13/G12, were examined and revealed no detectable pathogenic variants. The sole mutation identified was *EGFR* R252C. We have added this information in the revised manuscript and can be found at line 97.

4. There are some minor typos throughout the text. Please run a spell checker over the manuscript.

Response: We sincerely appreciate your careful reading. We have now thoroughly proofread the manuscript using both spell-check software and manual review to correct all typographical errors.

5. Figures 2a and 2b need quantification from three independent experiments.

Response: This point was well taken. We have now added quantitative analyses with statistical comparisons in the revised figure.

6. Is the Grb2-SOS-Kras axis not required for phosphorylation of ERKs? I wonder if a quick siRNA or inhibitor experiment could be done to verify that the direct binding of ERKs is sufficient for signalling by the R252C mutant.

Response: Thank you for the constructive comment and suggestion. To investigate whether *EGFR* R252C-induced ERK1/2 activation is dependent on Grb2-SOS-Kras, we treated U251 cells harboring either *EGFR* WT or *EGFR* R252C with Ras inhibitor (tipifarnib). As indicated by the result of Supplementary Fig. 2b, inhibition of RAS failed to block *EGFR* R252C-induced ERK1/2 activation, indicating that *EGFR* R252C-induced ERK1/2 activation is independent of Grb2-SOS-Kras pathway. The results were described in line 154.

7. Figure 3a: a negative control is missing.

Response: This point was well taken. We have repeated the experiment with the addition of a negative control. As shown in the new Fig. 3a, a similar result was obtained.

1 **Reviewer #1 (Remarks to the Author):**

2 Zhang et al. have addressed the major concerns from our first review and have added substantial
3 new data that strengthen the central claim that EGFR R252C directly drives oncogenic ERK
4 activation and is therapeutically targetable by afatinib. The revised manuscript and reviewer
5 response detail our requested mechanistic experiments (disulfide-dependent dimerization,
6 NanoBiT/crosslinking, P919 mutational tests), RAS/RAF/MEK inhibitor and genetic MEK
7 depletion controls, and side-by-side TKI comparisons and in vivo survival analyses. Overall,
8 the revisions are convincing and the authors should be commended for their effort and rigor
9 that went into addressing the comments. However, a few revisions/additions are requested to
10 further strengthen and clarify their conclusions.

11 **Response:** We thank the reviewer for the positive assessment and constructive suggestions. We
12 have conducted additional experiments and revised the manuscript accordingly. We believe
13 these changes have strengthened our conclusions. All changes we have made in the manuscript
14 have been highlighted in yellow.

15 1. Major: Provide final updated survival data and Kaplan-Meier analyses with the publication.
16 The survival endpoints of afatinib treatment strengthen the authors' translation claim. If data
17 remains censored, please provide censoring details and an explanation of why full curves could
18 not be presented.

19 **Response:** We thank the reviewer for the constructive suggestions. In Supplementary Fig. 6c,
20 the observation period was terminated upon the last death of mouse in the vehicle-treated
21 R252C group. At this endpoint, the surviving mice in other group remained in good condition,
22 with no significant weight loss or other adverse effects observed. These surviving mice were
23 right-censored to uphold ethical standards and statistical integrity, as the Kaplan-Meier method
24 appropriately handles censored data (PMID: 30015653, PMID:27713848, PMID: 21367692).
25 We have now added a detailed description of this approach in the Methods section (line 654
26 and line 672).

27 2. Minor: Clarify why afatinib shows apparent mutation-selective efficacy and fails to target
28 wild-type EGFR. Despite its known biochemical potency against wild-type EGFR (~0.5 nM;
29 Li et al., Oncogene 2008, in Supplemental Fig. 2a, afatinib does not show a significant impact
30 on pEGFR^{WT} and in Supplemental Fig. 5a neither afatinib nor other wild-type targeting EGFR
31 TKIs (erlotinib, gefitinib) significantly suppress ERK phosphorylation, even at high
32 concentrations (>1 μ M).

33 **Response:** Thank you for the comments. The additional experiment was conducted to
34 clarify the point. To activate EGFR, U251 cells were treated with EGF. As shown in new
35 Supplemental Fig. 5c, similar to erlotinib and gefitinib, afatinib effectively inhibited EGF-
36 induced phospho-ERK1/2 in EGFR WT cells upon EGF stimulation. The inhibitory efficacy of
37 afatinib was comparable between EGFR WT cells (with EGF stimulation) and EGFR R252C
38 mutant cells (Supplemental Fig. 5a and 5c). These results demonstrate that all three EGFR
39 inhibitors are functional in EGFR WT cells upon EGF stimulation. The results were described
40 in line 289.

41 However, as shown in Supplemental Fig. 5a, these inhibitors failed to suppress basal
42 phospho-ERK1/2 levels in U251 and H1299 cells. This may be because basal ERK1/2
43 activation in these cell lines is not primarily mediated by EGFR signaling.

44 Of note, the reported IC50 of ~0.5 nM (Li et al., 2008) was derived from an *in vitro* kinase
45 assay using purified EGFR protein. In contrast, our results are derived from cellular assays
46 conducted in U251 or H1299 cell lines. Such discrepancies in drug efficacy can be expected
47 when comparing biochemical and cellular systems.

48

49 **Reviewer #3 (Remarks to the Author):**

50 I would like to thank the authors for the additional data that has been provided and that
51 addresses some of the concerns that were raised. However, there are still some aspects that
52 require further clarification.

53 **Response:** We thank the reviewer for the positive assessment and constructive suggestions. We
54 have conducted additional experiments and revised the manuscript accordingly. We believe
55 these changes have strengthened our conclusions. All changes we have made in the manuscript
56 have been highlighted in yellow.

57 1. I find the legends in the supplementary figures too short to be able to understand what was
58 done. This makes it very difficult to understand the figures. For example, in Supplementary
59 figure 2: has this been done in full media with FBS, have the cells been stimulated, etc. In the
60 new supplementary figure 2, it is thus not obvious to understand whether the inhibitors actually
61 work. One would need to stimulate the cells with EGF in order to see whether the Ras, Raf and
62 MEK inhibitor actually work at the concentration used (what is the concentration anyway? –
63 not mentioned in the legend).

64 **Response:** We thank the reviewer for the constructive comments. We have now updated all
65 supplementary figure legends to explicitly describe the experimental conditions, such as the use
66 of full media and EGF stimulation. The concentrations of all inhibitors have also been clearly
67 indicated.

68 To validate inhibitor efficacy, we performed the additional experiment, in which U251
69 cells were serum-starved for 24 hr and then treated with RAS, RAF, or MEK inhibitors for 6 hr
70 prior to EGF stimulation for 30 min. The immunoblotting results, now included as
71 Supplemental Fig. 2e, confirm the effective inhibition of the target pathways. The results were
72 described in line 160.

73 2. I still do not see a reduced auto-phosphorylation for R252C. Supplementary Figure 4e does
74 not show this. In unstimulated cells, there is no phosphorylation visible in both wt and R252C
75 cells. Also, R252C should be shown with EGF stimulation. One cannot compare unstimulated
76 R252C to stimulated wt cells.

77 **Response:** As suggested by the reviewer, we assessed the phosphorylation status of key EGFR
78 tyrosine residues (Y845, Y992, Y1045, Y1086, Y1148, and Y1173) in U251 cells expressing
79 either EGFR WT or the R252C mutant, under both unstimulated and EGF-stimulated conditions.
80 Immunoblotting analysis revealed that upon EGF stimulation, cells expressing the EGFR
81 R252C mutant exhibited significantly reduced autophosphorylation levels at these tyrosine
82 residues compared to those expressing EGFR WT (Supplementary Fig. 4e and 4f). The results
83 were described in line 245.

REVIEWER#3

Well, I still have an issue with the auto-phosphorylation of the EGFR, but maybe this is just a misunderstanding. There is no auto-phosphorylation by wt nor the mutant receptor (i.e. phosphorylation without EGF stimulation, see supplementary figure 4e). But maybe what the authors mean to say is ABSENT auto-phosphorylation?

Response: We apologized for the confusion. Indeed, as shown in Supplementary Fig. 4e, no phosphorylated band was detected in the R252C group without EGF stimulation, indicating that the EGFR R252C mutant does not undergo auto-phosphorylation in the absence of EGF. This phenomenon is attributed to an increased distance between the kinase domains and an altered domain orientation in the R252C mutant. We have revised our manuscript to replace “reduced” with “absent”.